# Dynamic Pricing with Monotonicity Constraint Under Unknown Parametric Demand Model

**Su Jia, Andrew Li, R. Ravi**
Tepper School of Business
Carnegie Mellon University
{sjia1,aali1,ravi}@andrew.cmu.edu

## Abstract

We consider a Continuum-Armed Bandit problem with an additional monotonicity constraint (or "markdown" constraint) on the actions selected. This problem faithfully models a natural revenue management problem, called "markdown pricing", where the objective is to adaptively reduce the price over a finite horizon to maximize the expected revenues. Chen ([3]) and Jia et al ([9]) recently showed a tight $T^{3/4}$ regret bound over $T$ rounds under *minimal* assumptions of unimodality and Lipschitzness in the reward function. This bound shows that markdown pricing is strictly harder than unconstrained dynamic pricing (i.e., without the monotonicity constraint), which admits $T^{2/3}$ regret under the same assumptions ([11]). However, in practice, demand functions are usually assumed to have certain functional forms (e.g. linear or exponential), rendering the demand learning easier and suggesting better regret bounds. In this work we introduce a concept, *markdown dimension*, that measures the complexity of a parametric family, and present optimal regret bounds that improve upon the previous $T^{3/4}$ bound under this framework.

## 1 Introduction

Dynamic pricing under unknown demand arises naturally for the sale of new products, where the demand function is not available in advance. The seller in this case has to learn the demand function over time and hence faces a learning-vs-earning trade-off. This problem is therefore usually formulated as a Multi-Armed Bandit (MAB) model. Although bandit problems have been well understood *theoretically*, in practice, however, we rarely see retailers implement such policies. This is a largely because some practical constraints which can potentially cause customer dissatisfaction are overlooked. For example, a price increase may sometimes create a manipulative image of the retailer and negatively impact their ratings. For example, [13] analyzed the online menu prices of a set of restaurants and concluded that "on average, a 1% price increase leads to 3-5% decrease in online ratings". Therefore, retailers sometimes implicitly face a monotonicity constraint (which we call "markdown constraint"), which requires that the prices selected be non-increasing. A pricing policy that satisfies such a constraint is usually referred to as *markdown pricing* policy.

Thus motivated, in this work, we consider the markdown pricing problem with unknown demand, under various assumptions. Although *unconstrained* dynamic pricing under unknown demand has been extensively studied, little is known about *markdown* pricing under unknown demand. Recently, [3] and [9] independently showed that unimodality and Lipschitzness in the *revenue* function (defined as the product of the price and the mean demand) are necessary to achieve sublinear regret. In this setting, any reasonable policy must reduce the price at a moderate rate and stop only when there is sufficient evidence for *overshooting* the optimal price. By selecting suitable parameters, [3, 9] showed that the above policy is indeed optimal, achieving a tight $T^{3/4}$ regret bound.

36th Conference on Neural Information Processing Systems (NeurIPS 2022).

However, in practice, it is usually assumed that the demand function has a certain parametric form, such as linear, exponential or logit form. Intuitively, this extra structure enables faster learning rate and suggests better regret bounds. This motivates our first question:

Q1) Can we improve the $T^{3/4}$ regret bound for markdown pricing for parametric demand families?

Noticeably, there is a known *separation* result between under the assumption that the corresponding reward function is Lipschitz and unimodal. In fact, [3, 9] showed a tight $T^{3/4}$ regret bound, which is asymptotically higher than $T^{2/3}$, the optimal regret bound for *unconstrained* pricing ([11]) under the same assumptions. This highlights the extra complexity caused by the monotonicity constraint, and motivates the next question:

> Q2) Is there still a provable "separation" between markdown and unconstrained pricing under parametric assumptions?

While one can answer the Q1 and Q2 for specific parametric families, we aim at finding a unified framework that captures the hardness of demand learning for a given parametric family. This motivates the following question:

> Q3) Can we find a general framework to unify the regret bounds for different *categories* of families, rather than specific results for specific families?

In this work, we propose such a framework by introducing a concept called *markdown dimension*. We provide efficient markdown policies for each markdown dimension, which we also show to be *best* possible, thereby completely settling the problem.

## 1.1 Our Contributions.

In this work, we make the following contributions.

1. **New Complexity Measure for Demand Learning.** We introduce a new concept called *markdown dimension*, which captures the complexity of performing markdown pricing on a family, answering Q3. Within this framework, we provide a complete settlement of the problem as specified below.

2. **Markdown Policies with Theoretical Guarantees.** For each finite markdown dimension $d \geq 0$, we present an efficient markdown pricing policy with sublinear regret. The exploration proceeds in *phases* of varying lengths. In each phase $j$, the policies derive a confidence interval $I_j \subseteq I_{j-1}$ for the optimal price $p^*$ based on observations from evenly spaced prices, which are lower than the prices used in phase $j - 1$. The exploration terminates if there is sufficient evidence that the current price is lower than $p^*$. We show that for $d = 0$ and $d \geq 1$, our policies achieve regret $O(\log^2 T)$ and $\tilde{O}(T^{\frac{d}{d+1}})$ respectively, settling Q1.

3. **Separation From Unconstrained Pricing: Tight Lower Bounds.** We complement our upper bounds with a matching lower bound for each integer $d$. More precisely, we show an tight $\Omega(\log^2 T)$ lower bound for $d = 0$, which is lower than the known $O(\log T)$ regret bound without this monotonicity constraint (see [2]). For finite $d \geq 1$, we show an $\Omega(T^{\frac{d}{d+1}})$ lower bound, which not only matches our upper bound but is also higher than the tight $\tilde{\Theta}(T^{1/2})$ bound without markdown constraint (see [2, 10]). These lower bound results combined settle Q2.

4. **Impact of Smoothness:** We go further to refine our bounds and investigate the impact of smoothness of reward function around the optimal price. We consider a *sensitivity* parameter $s$, which essentially says that the reward function is only $O(\varepsilon^s)$ suboptimal if the price is at an $\varepsilon$ distance away from the optimal price. For both finite and infinite $d$, we extend our upper bounds to incorporate $s$.

As the most basic case, one can verify from Taylor's Theorem that when the demand function is assumed to admit (i) a continuous second derivative, and (ii) an optimal price in the interior of the domain, then $s = 2$. We highlight our results for $s = 2$ in red in Table 1, where $\Theta$ denotes tight upper and lower bounds.

| Markdown Dimension | Markdown Pricing | Unconstr. Pricing |
|---|---|---|
| $d = 0$ | $\Theta(\log^2 T)$ | $\Theta(\log T)$ [2] |
| $1 \leq d < \infty$ | $\tilde{\Theta}(T^{d/(d+1)})$ | $\tilde{\Theta}(\sqrt{T})$ [2] |
| $d = \infty$ | $\tilde{\Theta}(T^{3/4})$ [3, 9] | $\tilde{\Theta}(T^{2/3})$ [11] |

Table 1: Regret bounds for $s = 2$.

## 1.2 Related Work

In the *Multi-Armed Bandit* (MAB) problem, the player is offered a finite set of arms, with each arm providing a random revenue from an unknown probability distribution specific to that arm. The objective of the player is to maximize the total revenue earned by pulling a sequence of arms (e.g. [12]). Our pricing problem generalizes this framework by using an infinite action space $[0, 1]$ with each price $p$ corresponding to an action whose revenue is drawn from an unknown distribution with mean $R(p)$. In the *Lipschitz Bandit* problem (see, e.g., [1]), it is assumed that each $x \in [0, 1]$ corresponds to an arm with mean reward $\mu(x)$, where $\mu$ is an unknown $L$-Lipschitz function. For the one-dimensional case, [11] showed a tight $\tilde{\Theta}(T^{2/3})$ regret bound.

Recently there is an emerging line of work on bandits with monotonicity constraint. [3] and [9] recently independently considered the dynamic pricing problem with monotonicity constraint under unknown demand function, and proved nearly-optimal $T^{3/4}$ regret bound assuming the reward function is Lipschitz and unimodal. Moreover, they also showed that these assumptions are indeed minimal, in the sense that no markdown policy achieves $o(T)$ regret without any one of these two assumptions. Motivated by fairness constraints, [8] and [15] considered a general online convex optimization problem where the action sequence is required to be monotone.

Other constraints on the arm sequence motivated by practical problems are also considered in the literature. For example, motivated by the concern that customers are hostile to frequent price changes (see e.g. [5]), [4, 14] and [16] considered pricing policies with a given budget in the number of price changes . Motivated by clinical trials, [6] studied the best arm identification problem when the reward of the arms are monotone and the goal is to identify the arm closest to a threshold.

## 2 Model and Assumptions

We begin by formally stating our model. In this work we assume an unlimited supply of a single product. Given a discrete time horizon of $T$ rounds, in each round $t$, the policy (representing the "seller") selects a price $p_t$ (the particular interval $[0, 1]$ is without loss of generality, by scaling). The demand $D_t$ in this round is then independently drawn from a fixed distribution with *unknown* mean $D(p_t)$, and the policy receives revenue (or *reward*, which we will use interchangeably) $p_t$ for each unit sold, and hence a total of $p_t \cdot D_t$ revenue in this round. The only constraint the policy must satisfy is the *markdown* constraint: $p_1 \geq \cdots \geq p_T$ with probability 1.

The function $D(p)$ which maps each price $p$ to the mean demand at this price is known as the *demand function*. For any policy[1] $\pi$ and demand function $D(\cdot)$, we use $r(\pi, D)$ to denote the expected total reward of $\pi$ under $D$. Rather than evaluating policies directly in terms of $r(\pi, D)$, it is more informative (and ubiquitous in the literature on MAB) to measure performance using the notion of *regret* with respect to a certain idealized benchmark. Specifically, since we assumed unlimited supply, when the true reward function is known, the seller simply always selects a revenue-maximizing price $p_D^* = \arg \max_{p \in [0,1]} p \cdot D(p)$ at each round, and we denote this maximal reward rate to be $r_D^* = \max_{p \in [0,1]} p \cdot D(p)$. The regret of a policy is then defined with respect to this quantity, and we seek to bound the *worst-case* value over a given family of demand functions.

**Definition 1** (Regret). For any policy $\pi$ and demand function $D$, define the *regret* of policy $\pi$ under $D$ as $\text{Reg}(\pi, D) := r_D^* \cdot T - r(\pi, D)$. For any given family $\mathcal{F}$ of demand functions, the *worst-case regret* (or simply *regret*) of policy $\pi$ for family $\mathcal{F}$ is $\text{Reg}(\pi, \mathcal{F}) := \sup_{D \in \mathcal{F}} \text{Reg}(\pi, D)$.

---

[1]Formally, a *policy* is a sequence $\pi = (\pi_t)_{t \in [T]}$ of mappings where $\pi_t : [0, 1]^{t-1} \times \mathbb{R}_{>0}^{t-1} \to [0, 1]$ corresponds to the decision made at time $t$, based on the realized demands and prices selected up till time $t - 1$.

## 2.1 Basic Assumptions

Now we state the common assumptions that all of our results rely on. A demand function $D(\cdot)$ is associated with a *revenue function* (or, *reward function*) $R(p) = p \cdot D(p)$. Sometimes it is more convenient to work directly with the revenue functions, in which cases we abuse the notations and denote by $\text{Reg}(\pi, R)$ the regret of under reward function $R$.

**Definition 2** (Optimal Price Mapping). Let $\mathcal{F}$ be a set of functions defined on $S \subseteq \mathbb{R}$. For any function $R : S \to \mathbb{R}$, let $M(R)$ be the subset of its global maxima on $S$. The *optimal price mapping* $p^* : \mathcal{F} \to S$ is defined as $p^*(R) = \inf M(R)$.

We first introduce a standard assumption (see e.g. [2]) that the derivative of $R$ vanishes at $p^*(R)$.

**Assumption 1** (First Order Optimality). We assume that every reward function $R$ is differentiable on its domain, and moreover, $R'(p^*(R)) = 0$.

If, in addition, $R$ is twice continuously-differentiable, then by Taylor expansion around $p^*$, $|R(p^*) - R(p^* + \varepsilon)| = \frac{1}{2} R''(p^*) \cdot \varepsilon^2 + o(\varepsilon^2)$. Thus, if a policy overshoots the optimal price by $\varepsilon$, then only an $O(\varepsilon^2)$ loss is incurred in each round. We next introduce a *distributional* assumption.

**Definition 3** (Subgaussian Random Variable). Define the *subgaussian norm* of a random variable $X$ as $\|X\|_{\psi_2} := \inf\{c > 0 : \mathbb{E}[e^{X^2/c^2}] \leq 2\}$. Say $X$ is *subgaussian* if $\|X\|_{\psi_2} < \infty$.

**Assumption 2** (Subgaussian noise). There exists a constant $C_{sg} > 0$ such that under any true demand function and any price $p$, the random demand $X$ at price $p$ satisfies $\|X\|_{\psi_2} \leq C_{sg}$.

## 2.2 Warm-up: Markdown Pricing on Linear Demand

To introduce the key concept, markdown dimension, let's take linear demand as a warm-up. Consider $D(x; \theta) = \theta_1 - \theta_2 x$ for $x \in \left[\frac{1}{2}, 1\right]$ and the following natural markdown policy for $\mathcal{F} = \left\{ D(x; \theta) : \theta_1, \theta_2 \in \left[\frac{1}{2}, 1\right] \right\}$: collect a number of samples at two prices $p_1, p_2$ close to 1, and denote by $\bar{d}_1, \bar{d}_2$ the empirical mean demands. By the Fundamental Theorem of Algebra, since a degree-$d$ polynomial can be uniquely determined by its value at $d$ distinct points, there is a $\hat{\theta} \in \mathbb{R}^2$ satisfying $D(p_i; \hat{\theta}) = \bar{d}_i$ for $i = 1, 2$. Finally, select the optimal price of $D(\cdot; \hat{\theta})$ in remaining rounds.

This policy is "robust" in the following sense. Suppose $\bar{d}_i$ deviates from the mean $D(p_i; \theta)$ by $\sim \delta$, then the estimation error in $\theta$ is $\sim \frac{\delta}{|p_1 - p_2|}$, i.e., proportional to $\delta$ and inverse proportional to the gap between the two sample prices.

For more complex demand models, the dependence on the "divergence" of the sample prices changes. To characterize such dependence, we introduce the notion of *markdown dimension*. Informally, a parameterized family has markdown dimension $d$ if for any set of $d$ prices at distance $h$ apart, an $O(\delta)$ error on the estimated demands results in an $O\left(\frac{\delta}{h^d}\right)$ estimation error on the model parameter. We make this idea formal in the subsequent sections.

## 2.3 Identifiability and Robust Parametrization

Intuitively, the exploration-exploitation trade-off for markdown pricing becomes harder to manage as the given family becomes more complex. Consider, for example, linear demand functions. If each function takes the form $D(p; c) = 1 + cp$ where only $c \in (0, 1)$ is unknown and $p \in [0, 1]$, then the seller simply needs to estimate the price elasticity $c$ by sampling sufficiently many times at $p = 1$.

In contrast, if the demand function is $D(p; a, b) = a - bp$ where *both* $a, b$ are unknown, then sampling at one price would *not* suffice. Rather, one needs to select (at least) two distinct prices, say $p$ and $p'$, to fit a linear function, thereby facing the following dilemma. Suppose $p < p'$ are far apart, then, $p$ may be far away from the optimal price $p^*$ if $p^*$ is close to $p'$, resulting in high regret. Otherwise, when $p, p'$ are close-by, the demand learning requires a high volume of samples, which also leads to a high regret.

Thus we reach a natural question: can we introduce a notion of complexity to measure the difficulty of performing markdown pricing on a given family, and then provide tight regret bounds in terms for each level of complexity? In this work, we propose a novel concept of *markdown dimension* and provide nearly-optimal regret bounds in terms of this complexity metric. The formal definition relies

on two other concepts which we introduce in the next two subsections: the *identifiability* of a family, and the *robustness* of a parametrization.

**Identifiability** Our notion of identifiability generalizes a key property of single-variable polynomials, that every degree-$d$ polynomial can be uniquely determined by its values at *any* $(d+1)$ points. To present the formal definition, we first introduce a mapping which, for a fixed subset of prices, assigns each demand function a *profile* based on its values at those prices.

**Definition 4** (Profile Mapping). Consider a set $\mathcal{F}$ of real-valued functions defined on $A \subseteq \mathbb{R}$. For any $\mathbf{p} = (p_0, p_1, ..., p_d) \in A^{d+1}$, the *profile-mapping* at prices $\mathbf{p}$ is defined as

$$\Phi_{\mathbf{p}} : \mathcal{F} \to \mathbb{R}^{d+1},$$
$$D \mapsto (D(p_0), D(p_1), ..., D(p_d)).$$

We may subsequently call $\Phi_{\mathbf{p}}(D)$ the *profile* of function $D$ at $\mathbf{p}$.

Informally, a family of functions is said to be $d$-identifiable, if the graphs of any two functions intersect for at most $d$ times. For example, the family of all degree $d$ polynomials is $d$-identifiable.

**Definition 5** (Identifiability). A family $\mathcal{F}$ of functions defined on $S \subseteq \mathbb{R}$ is *$d$-identifiable*, if for any *distinct* $p_0, p_1..., p_d \in S$, the profile mapping $\Phi_{p_0,...,p_d}$ is injective, i.e. $\mathcal{F}(f) \neq \mathcal{F}(\tilde{f})$ for any distinct $f, \tilde{f} \in \mathcal{F}$.

We will soon use the following fact: if a family is $d$-identifiable, then for any distinct $p_0, p_1..., p_d$ the inverse profile-mapping $\Phi_{\mathbf{p}}^{-1} : \mathcal{R}_p \to \mathcal{F}$ exists, where $\mathcal{R}_p$ is the range of the mapping $\Phi_{\mathbf{p}}$.

**Robust Parametrization** We first formally define a *parametrization*.

**Definition 6** (Parametrization). An *order-$m$ parametrization* for a family $\mathcal{F}$ of functions is a one-to-one mapping from some set $\Theta \subseteq \mathbb{R}^m$ to $\mathcal{F}$. Moreover, each $\theta \in \Theta$ is called a *parameter*.

We use $D(p; \theta)$ to denote the function in $\mathcal{F}$ that parameter $\theta$ corresponds to. For example, for $D(p; \theta_1, \theta_2) = \theta_1 - \theta_2 \cdot p$ is a parametrization of the family of linear functions. As a standard assumption (see e.g. [2]), we assume $\Theta$ to be compact, which leads to many favorable properties.

**Assumption 3** (Compact Domain). The domain $\Theta$ of the parametrization is compact.

Under this assumption, the demand functions in $\mathcal{F}$ are bounded, and thus we may without loss of generality scale the range (i.e. target space) of those functions to be $[0, 1]$.

**Assumption 4** (Smoothness). The mapping $D : [0, 1] \times \Theta \to \mathbb{R}$ is twice continuously differentiable. In particular, since $[0, 1] \times \Theta$ is compact, under the above assumption, there exist constants $C^{(j)} > 0$ such that $|D^{(j)}(p, \theta)| \leq C^{(j)}$ for any $(p, \theta) \in [0, 1] \times \Theta$ and $j = 0, 1, 2$.

By abuse of notation, we view the optimal price mapping $p^*$ as being defined on $\Theta$, rather than on $\mathcal{F}$ as before. Formally, let $M(\theta)$ be the set of global maxima of $R(x; \theta)$, then $p^*(\theta) = \inf R(x; \theta)$. For example, for $D(p; \theta_1, \theta_2) = \theta_1 - \theta_2 \cdot p$ one can verify that $p^* = \frac{\theta_1}{2\theta_2}$. The next assumption allows us to "propagate" the estimation error in $\theta$ to that of $p^*(\theta)$.

**Assumption 5** (Lipschitz Optimal Price Mapping). The optimal price mapping $p^* : \Theta \to [0, 1]$ is $C^*$-Lipschitz for some constant $C^* > 0$.

This assumption has appeared in the previous literature on parametric demand learning, see e.g. Assumption 1(c) of [2]. Moreover, it is satisfied by many basic demand functions such as linear, exponential and logit demand. For instance, let $D(p; c) = 1 - cp$ where $p, c \in [0, 1]$, then $p^*(c) = \min\{\frac{1}{2c}, 1\}$, which is 1-Lipschitz. Nonetheless, this assumption is somewhat unnatural as it is not made on the parametrization directly, we leave it open whether there is an assumption imposed directly on $\mathcal{F}$ that implies this assumption.

The final ingredient for robust parametrization is motivated by a nice property of the *natural parametrization* $D(p; \theta) = \sum_{j=0}^{d} \theta_j p^j$ for polynomials. Consider any distinct $p_0, p_1, ..., p_d$, and distinct real numbers $y_0, y_1, ..., y_d$ representing, for example, the mean reward at each $p_i$. We can then uniquely determine a degree-$d$ polynomial by solving the linear equation $V_{\mathbf{p}}\theta = \mathbf{y}$ where

$\theta = (\theta_0, \theta_1 ..., \theta_d)^T$, $\mathbf{y} = (y_0, y_1, ..., y_d)^T$ and $V_\mathbf{p} := V(p_0, ..., p_d)$ is the *Vandermonde* matrix. One can easily verify that $V_\mathbf{p}$ is invertible if and only if $p_i$'s are distinct, in which case we have $\theta = V_\mathbf{p}^{-1} \mathbf{y}$. Next we consider the effect of a perturbation on $\mathbf{y}$, in terms of the following *separability* parameter.

**Definition 7** (Separability)**.** For any $\mathbf{p} = (p_0, ..., p_d) \in \mathbb{R}^{d+1}$, define $h(\mathbf{p}) := \min_{i \neq j} |p_i - p_j|$.

To motivate the general definition of robustness, first consider a result specific to polynomials. Recall that $\mathcal{R}_\mathbf{p}$ is range of the profile mapping $\Phi_\mathbf{p}$.

**Proposition 1** (The Natural Parametrization for Polynomials is Robust)**.** There exist constants $C_1, C_2 > 0$ such that for any $\mathbf{p} \in [0, 1]^{d+1}$ with $0 < h(\mathbf{p}) \leq C_1$, and $\mathbf{y}, \hat{\mathbf{y}} \in \mathcal{R}_\mathbf{p}$ with $\|\mathbf{y} - \hat{\mathbf{y}}\|_\infty \leq C_1$, it holds that $\|V_\mathbf{p}^{-1} \mathbf{y} - V_\mathbf{p}^{-1} \hat{\mathbf{y}}\|_\infty \leq C_2 \cdot \|\mathbf{y} - \hat{\mathbf{y}}\|_\infty \cdot h(\mathbf{p})^{-d}$.

More concretely, let $D(p; \theta)$ be the underlying polynomial demand function, then $\mathbf{y} = V_\mathbf{p}\theta$ are the mean demands at the prices in $\mathbf{p}$. Suppose we observe empirical mean demands $\hat{\mathbf{y}} = (\hat{y}_0, \hat{y}_1, ..., \hat{y}_d)$ at prices $\mathbf{p}$, then we have a reasonable estimation $\hat{\theta} = V_\mathbf{p}^{-1} \hat{\mathbf{y}}$. Our Proposition 1 can then be viewed as an upper bound on the estimation error $\|\theta - \hat{\theta}\|_\infty$ in terms of $h(\mathbf{p})$ and $\mathbf{y} - \hat{\mathbf{y}}$.

In order to achieve sublinear regret, we need to ensure $h(\mathbf{p}) = o(1)$ as $T \to \infty$. Moreover, the rate of this convergence crucially affects our regret bounds. Proposition 1 establishes a nice property for polynomials, that the estimation error scales as $h^{-d}$, as $h \to 0^+$. We introduce *robust parametrization* by generalizing this property beyond polynomials. Loosely, an order-$d$ parametrization is *robust*, if the bound in Proposition 1 holds.

**Definition 8** (Robust Parametrization)**.** An order-$d$ parametrization $\theta : \Theta \to \mathcal{F}$ is *robust*, if
(1) it satisfies Assumptions 3, 4 and 5, and
(2) there exist constants $C_1, C_2 > 0$ such that for any $\mathbf{p} \in \mathbb{R}^{d+1}$ with $0 < h(\mathbf{p}) \leq C_1$ and any $y, y' \in \mathcal{R}_\mathbf{p}$ with $\|\mathbf{y} - \mathbf{y}'\|_\infty \leq C_1$, it holds that $\|\Phi_\mathbf{p}^{-1}(\mathbf{y}) - \Phi_\mathbf{p}^{-1}(\mathbf{y}')\|_\infty \leq C_2 \cdot \|\mathbf{y} - \mathbf{y}'\|_\infty \cdot h(\mathbf{p})^{-d}$.
In particular, when $d = 0$, this inequality simply says $\Phi_\mathbf{p}^{-1}$ is $C_2$-Lipschitz.

## 2.4   Markdown Dimension

Now we are ready to define the markdown dimension.

**Definition 9** (Markdown Dimension)**.** The *markdown dimension* (or simply *dimension*) for a family $\mathcal{F}$ of functions, denoted $d(\mathcal{F})$, is the minimum integer $d \geq 0$ such that $\mathcal{F}$ is (i) $d$-identifiable, and (ii) admits a robust order-$d$ parametrization. If no finite $d$ satisfies the above conditions, then $d(\mathcal{F}) = \infty$.

We further illustrate our definition by considering the dimensions of some basic families. As the simplest family, one may verify that our definition of 0-dimensional family is equivalent to the *separable* family as defined in Section 4 of [2]. We provide more concrete examples below. Details can found in Appendix A.

**Proposition 2.** Let $D(x; \theta) = \sum_{j=0}^d \theta_j x^j$ and suppose $\mathcal{F} = \{D(x; \theta) : \theta \in \Theta\}$ satisfies Assumptions 3, 4 and 5. Then, $d(\mathcal{F}) = d$.

**Proposition 3** (0-Dimensional Families)**.** The following families $\mathcal{F}_1, \mathcal{F}_2$ and $\mathcal{F}_3$ are 0-dimensional.
(1) **[Single-Parameter Linear Demand]** Let $D_a(x) = 1 - ax$ for $x \in \left[\frac{1}{2}, 1\right]$ and $\mathcal{F}_1 = \{D_a(x) : a \in \left[\frac{1}{2}, 1\right]\}$.
(2) **[Exponential Demand]** Let $D_a(x) = e^{1-ax}$ for $x \in \left[\frac{1}{2}, 1\right]$ and $\mathcal{F}_2 = \{D_a(x) : a \in \left[\frac{1}{2}, \frac{3}{4}\right]\}$.
(3) **[Logit Demand]** Let $D_a(x) = \frac{e^{1-ax}}{1+e^{1-ax}}$ for $x \in \left[\frac{1}{2}, 1\right]$ and $\mathcal{F}_3 = \{D_a(x) : a \in \left[\frac{1}{2}, 1\right]\}$.

Finally, we observe that by our definition, if a family of functions is not $d$-identifiable for any $d$, then $d(\mathcal{F}) = \infty$. For example, for any $d$ distinct prices, there exist multiple (more precisely, infinitely many) Lipschitz functions having the same values on these $d$ prices.

**Proposition 4.** Let $\mathcal{F}$ be the set of 1-Lipschitz functions from $[0, 1]$ to $[0, 1]$, then $d(\mathcal{F}) = \infty$.

## 2.5   Sensitivity

Consider the Taylor expansion of a reward function $R(x)$ around an optimal price $p^*$: $R(p) = R(p^*) + 0 + \frac{1}{2!} R''(p^*)(p - p^*)^2 + \frac{1}{3!} R^{(3)}(p^*)(p - p^*)^3 + ...$ Suppose the first nonzero derivative

is $R^{(k)}(p^*)$. Then, the higher $k$, the less *sensitive* the revenue is to the estimation error in $p^*$. We capture this aspect in the following notion of *sensitivity*.

**Definition 10** (Sensitivity). A reward function $R$ is *s-sensitive* if it is $(s+1)$-times continuously differentiable with $R^{(1)}(p^*(R)) = \cdots = R^{(s-1)}(p^*(R)) = 0$ and $R^{(s)}(p^*(R)) < 0$. A family $\mathcal{F}$ of reward functions is called *s-sensitive* if
(a) every $R \in \mathcal{F}$ is $s$-sensitive,
(b) it admits a parametrization $R(x;\theta)$ satisfying Assumptions 3 to 5,
(c) there is a constant $C_6 > 0$ such that $R^{(s)}(p^*(R)) \leq -C_6 < 0$ for any $R \in \mathcal{F}$, and
(d) for each $0 \leq j \leq s$, there exists a constant $C^{(j)}$ s.t. $|R^{(j)}(x;\theta)| \leq C^{(j)}, \forall x \in [0,1], \theta \in \Theta$.

For example, let $s \geq 3$ and $R(x;\theta) = \theta - |\frac{1}{2} - x|^s$ for $x \in [0,1]$ and any $\theta \in \mathbb{R}$, then $\{R(x;\theta) : \theta \in [\frac{1}{2}, 1]\}$ is an $s$-sensitive family. Note that by Taylor's Theorem, $|R(p^* + \varepsilon) - R(p^*)| \leq O(\varepsilon^s)$. Consequently, if a policy errs by $\varepsilon$ in the optimal price, the regret *per round* is $O(\varepsilon^s)$, lower than the per-round regret $O(\varepsilon^2)$ in the basic case. In each of the next three sections, we will first present the regret bounds for the basic case $s = 2$, and then characterize how this bound improves as $s$ increases.

## 3 Zero-Dimensional Family

We start with the simplest case, $0$-dimensional demand functions. As opposed to the *optimism* in the face of uncertainty in UCB policies, our *Cautious Myopic* policy (Algorithm 1) adopts *conservatism* in the face of uncertainty. More precisely, we partition the time horizon into phases where *phase* $j$ consists of $t_j = \lceil 9^j \log T \rceil$ rounds, and thus in total there are $K = O(\log T - \log \log T)$ phases. In each phase, the policy builds a confidence interval $I_j$ for the true parameter using the samples collected in this phase, and sets the price for phase $j + 1$ to be the *largest* optimal price of any "surviving" parameter $\theta \in I_j$. We write $t^{(j)} := \sum_{k=0}^{j} t_k$ and for convenience $t^{(0)} = 0$.

---

**Algorithm 1** Cautious Myopic (CM) Policy.

---

1: Input: a family $\mathcal{F}$ of demand functions and time horizon $T$.
2: $p_1 \leftarrow 1$                % Initialization
3: **for** $j = 1, ..., K$ **do**
4:   **for** $t = t^{(j-1)} + 1, \cdots, t^{(j-1)} + t_j$ **do**
5:    $x_t \leftarrow p_j$          % Select $p_j$ for $t_j$ times in phase $j$
6:    Observe realized demand $D_t$
7:   **end for**
8:   $\bar{d}_j = \frac{1}{t_j} \sum_{\tau=1}^{t_j} D_{t^{(j-1)}+\tau}$      % Empirical mean demand in phase $j$
9:   $\hat{\theta}_j \leftarrow \Phi_{p_j}^{-1}(\bar{d}_j)$         % Estimate parameter
10:   $w_j \leftarrow 4C_2 \cdot C_{sg} \sqrt{\frac{\log T}{t_j}}$      % Width of the confidence interval
11:   $\tilde{p}_{j+1} \leftarrow \max \left\{ p^*(\theta) : |\theta - \hat{\theta}_j| \leq w_j \right\}$   % Conservative estimation of the optimal price
12:   $p_{j+1} \leftarrow \min \{\tilde{p}_{j+1}, p_j\}$     % Ensure monotonicity
13: **end for**

---

**Theorem 1** (Zero-Dimensional Upper Bound). Let $\mathcal{F}$ be any $0$-dimensional, $s$-sensitive family of demand functions. Then the Cautious Myopic (CM) Policy has regret

$$\text{Reg}(\text{CM}, \mathcal{F}) = \begin{cases} O(\log^2 T), & \text{if } s = 2, \\ O(\log T), & \text{if } s > 2. \end{cases}$$

**Remark 1.** It is worth noting that this bound is asymptotically higher than the $O(\log T)$ upper bound for unconstrained pricing ([2]). Intuitively, this is because the CM policy purposely makes conservative choices of prices to avoid overshooting.

But can we achieve $o(\log^2 T)$ regret by behaving less conservatively? We answer this question negatively, and provide a *separation* between markdown and unconstrained pricing for dimension $0$.

**Theorem 2** (Zero-Dimensional Lower Bound). There exists a $0$-dimensional $2$-sensitive family $\mathcal{F}$ such that for any policy $\pi$, we have $\text{Reg}(\pi, \mathcal{F}) = \Omega(\log^2 T)$.

# 4 Finite-Dimensional Family

For dimension $d$, the learner needs $d + 1$ distinct *sample prices*, as opposed to just one price when $d = 0$. This, however, incurs extra regret, since the optimal price may lie *between* these sample prices. A policy faces the following trade off. If the gap is large, the policy may learn the parameter efficiently, but there is potentially a higher regret due to overshooting, if the true optimal price lies between the sample prices. On the other extreme, if the gap is small, there is less risk of overshooting but a slower rate of learning.

We introduce our Iterative Cautious Myopic (ICM) Policy (Algorithm 2) that hits the sweet spot. The policy consists of $m$ phases. In phase $j \in [m]$, it selects $p_j, p_j - h, ..., p_j - dh$ each for $T_j$ times. Based on the observed demands at these prices, the policy computes an estimated optimal price $\hat{p}_j$ and a confidence interval $[L_j, U_j]$. To determine the initial price $p_{j+1}$ of the next phase, we consider the relationship between the confidence interval $[L_j, U_j]$ and $p_j - dh$. There are three cases:
1. **Good Event.** If $p_j - dh > U_j$, then we get closer to the optimal price by setting $p_{j+1} = U_j$.
2. **Dangerous Event.** If $L_j \leq p_j - dh \leq U_j$, i.e. the current price is already within the confidence interval, the policy needs to behave conservatively by setting $p_{j+1} = p_j - dh$ to prevent from overshooting further.
3. **Overshooting Event.** If $L_j > p_j - dh$, i.e., the current price has already "overshot", the policy exits the exploration phase and selects the current price in the remaining rounds.

---

**Algorithm 2** Iterative Cautious Myopic (ICM) Policy.

---

1: Input: $\mathcal{F}, m, \{T_j\}_{j \in [m]}, T$
2: $p_1 \leftarrow 1, L_0 \leftarrow 0, U_0 \leftarrow 1$ Initialization
3: **for** $j = 1, 2, ...m$ **do**
4:     **for** $k = 0, 1, ..., d$ **do**
5:         Select price $p_j - kh$ for $T_j$ times and observe $D_1, ..., D_{T_j}$
6:         $\bar{D}_k \leftarrow \frac{1}{T_j} \sum_{i=1}^{T_j} D_i$               % Mean demand at $p_j - kh$
7:     **end for**
8:     $\hat{\theta} \leftarrow \Phi_{p_j, ..., p_j - dh}^{-1} \left( \bar{D}_0, ..., \bar{D}_d \right)$           % Estimate the parameter
9:     $w_j \leftarrow 2h^{-d} \cdot C_2 \cdot C_{sg} \sqrt{\frac{d \log T}{T_j}}$       % Width of confidence interval
10:    $L_j \leftarrow \min \left\{ p^*(\theta) : \left\| \theta - p^* \left( \hat{\theta} \right) \right\|_2 \leq w_j \right\}$      % Lower confidence bound
11:    $U_j \leftarrow \max \left\{ p^*(\theta) : \left\| \theta - p^* \left( \hat{\theta} \right) \right\|_2 \leq w_j \right\}$      % Upper confidence bound
12:    **if** $U_j > p_j - dh$ **then**
13:       $p_{j+1} \leftarrow U_j$                     % Good event
14:    **end if**
15:    **if** $U_j \geq p_j - dh \geq L_j$ **then**
16:       $p_{j+1} \leftarrow p_j - dh$              % Dangerous event
17:    **end if**
18:    **if** $p_j - dh < L_j$ **then**
19:       Break                    % Overshooting event
20:    **end if**
21: **end for**
22: Select the current price in every future round          % Exploitation

---

**Theorem 3** (Upper Bound for Finite $d \geq 1$). Suppose $s = 2$, then for $h = T^{\frac{m}{m(d+1)+1}}$, $T_j = T^{\frac{md+j}{m(d+1)+1}}$ and $m = \log T$, we have $\text{Reg}(\text{ICM}, \mathcal{F}) = \tilde{O}\left( T^{\frac{d}{d+1}} \right)$. More generally, for any $s \geq 2$ and $m = \tilde{O}(1)$, there exist $T_1 < ... < T_m$ such that $\text{Reg}(\text{ICM}, \mathcal{F}) = \tilde{O}\left( T^{\rho(m,s,d)} \right)$ where

$$\rho(m, s, d) = \frac{1 + \left( 1 + \frac{s}{2} + ... + \left( \frac{s}{2} \right)^{m-1} \right) d}{\left( \frac{s}{2} \right)^m + \left( 1 + \frac{s}{2} + ... + \left( \frac{s}{2} \right)^{m-1} \right) \cdot (d+1)}.$$

We complement the upper bound with a nearly tight lower bound for $s = 2$, as stated below.

**Theorem 4** (Lower Bound for Finite $d \geq 1$). For any $d \geq 2$, there exists a $d$-dimensional family $\mathcal{F}$ of demand functions such that for any markdown policy $\pi$, we have $\text{Reg}(\pi, \mathcal{F}) = \Omega(T^{\frac{d}{d+1}})$.

In our proof, for each $d \geq 1$ we construct a family of $(d+1)$-degree, decreasing polynomial functions – which is also $d$-dimensional – on which any policy suffers regret $\Omega(T^{\frac{d}{d+1}})$.

**Remark 2.** At first glance, when $d \geq 4$ this result seems to contradict the $T^{3/4}$ upper bound for unimodal, Lipschitz reward family ([3, 9]). This is not a contradiction because the unimodality assumption no longer holds. In fact, a degree-$d$ polynomial with $d \geq 4$ is in general not unimodal.

**Remark 3.** An interesting case is $d = 1$. In this case, the revenue function is unimodal, so the $T^{\frac{3}{4}}$ upper bound from [3, 9] applies, but Theorem 3 gives a stronger $T^{\frac{1}{2}}$ upper bound.

We conclude the section by pointing out a limitation of our results. For each $d \geq 1$, our regret bounds is tight in the *exponent* of $T$, however, the constants in the big-O have different dependence on $d$. In fact, the upper and lower bound become exponentially higher and lower in $d$ respectively as $d$ grows.

# 5   Infinite Dimensional Family

When $d = \infty$, it is more convenient to work with the reward function $R(x) = x \cdot D(x)$ instead of the demand function. In contrast to *unconstrained* pricing, there is no *markdown* policy that achieves $o(T)$ regret on the family of Lipschitz reward functions ([3, 9]), since the reward function may have multiple local maxima. Nonetheless, [3] and [9] showed that if in addition we assume the reward function to be unimodal, then the optimal regret is $T^{3/4}$. Specifically, the lower bound is established by considering reward functions that may change *abruptly* at $p^*$. We next show that the regret bound can be improved if we instead assume the reward function changes smoothly. In spirit, our policy is similar to that of [3] and [9]: reduce the price at uniform speed and terminate when there is sufficient evidence for "overshooting", but we will achieve improved guarantee by choosing a different step size $\Delta$ that decreases in the smoothness $s$.

---

**Algorithm 3** Uniform Elimination Policy (UE$_{m,\Delta}$).

---

1: Input: $T, \Delta, m > 0$
2: Initialize: $L_{\max} \leftarrow 0$, $w \leftarrow 2C_{sg}\sqrt{\frac{\log T}{m}}$
3: **for** $j = 0, 1, 2, ..., \lceil \Delta^{-1} \rceil$ **do**
4:     $x_j \leftarrow 1 - j\Delta$
5:     Select price $x_j$ for the next $m$ rounds and observe rewards $Z_1^j, ..., Z_m^j$
6:     $\bar{\mu}_j \leftarrow \frac{1}{m}\sum_{i=1}^m Z_i^j$                                    % Empirical mean reward
7:     $[L_j, U_j] \leftarrow [\bar{\mu}_j - w, \bar{\mu} + w]$                            % Compute confidence interval
8:     **if** $L_j > L_{\max}$ **then**
9:         $L_{\max} \leftarrow L_j$                                                    % Keep track of the highest $L_j$
10:     **end if**
11:     **if** $U_j < L_{\max}$ **then**
12:         $h \leftarrow j$                                    % Exploration ends, define the *halting price* $x_h$
13:         Break
14:     **end if**
15: **end for**
16: Select price $x_h$ in all remaining rounds.                            % Exploitation phase

---

**Theorem 5** (Upper Bound for Infinite-Dimensional Family). Let $\mathcal{F}_s^U$ be the family of unimodal, $s$-sensitive reward functions. For any $s \geq 2$, there exist $m, \Delta$ with $\text{Reg}(\text{UE}_{m,\Delta}, \mathcal{F}_s^U) = O(T^{\frac{2s+1}{3s+1}})$.

We complement the above upper bound with a matching lower bound for every $s \geq 2$.

**Theorem 6** (Lower Bound for $s$-Sensitive Family). For any $s \geq 2$, there is a family $\mathcal{F}$ of $s$-sensitive unimodal revenue functions such that any markdown policy $\pi$ satisfies $\text{Reg}(\pi, \mathcal{F}) = \Omega(T^{\frac{2s+1}{3s+1}})$.

**Remark 4.** As a final remark, we point out that all our results assume $d$ is given. If instead the learner only has knows an upper bound $d_{\max}$ on $d$, then she can simply choose the policy for $d_{\max}$. We leave it open whether a best-of-all-worlds algorithm exists.

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
