# Dynamic Pricing with Monotonicity Constraint Under Unknown Parametric Demand Model

## Abstract

We consider a Continuum-Armed Bandit problem with an additional monotonicity constraint (or "markdown" constraint) on the actions selected. This problem faithfully models a natural revenue management problem, called "markdown pricing", where the objective is to adaptively reduce the price over a finite sales horizon to maximize expected revenues. Chen ([4]) and Jia et al ([9]) recently showed a tight $T^{3/4}$ regret bound over $T$ rounds under *minimal* assumptions of unimodality and Lipschitzness in the reward function. This bound shows that markdown pricing is strictly harder than unconstrained dynamic pricing (i.e., without the monotonicity constraint), which only suffers regret $T^{2/3}$ under the same assumptions ([11]). However, in practice, demand functions are usually assumed to have certain functional forms (e.g. linear or exponential), rendering the demand learning easier and suggesting lower regret bounds. We introduce a concept, *markdown dimension*, that measures the complexity of any parametric family, and present optimal regret bounds that improve upon the previous $T^{3/4}$ bound under this framework.

## 1 Introduction

Dynamic pricing under unknown demand arises naturally for the sale of new products, where the demand function is not available in advance. The seller in this case has to learn the demand function over time and faces a learning-vs-earning trade-off. This problem is therefore usually formulated as a Multi-Armed Bandit (MAB) problem. Although bandit problems have been well understood *theoretically*, in practice, however, we rarely see retailers implement such policies. This is a largely because some practical constraints are often overlooked by those policies. For example, price increases can potentially create a manipulative image of the retailer and negatively impact their ratings. For example, for online menu prices, "on average, a 1% price increase leads to 3-5% decrease in online ratings" ([13]). Therefore, retailers may sometimes implicitly face a monotonicity constraint (which we call "markdown constraint"), which requires that the prices selected be non-increasing. A pricing policy that satisfies such a constraint is usually referred to as *markdown pricing* policy.

Thus motivated, in this work, we consider the markdown pricing problem with unknown demand, under various assumptions. Although *unconstrained* dynamic pricing under unknown demand has been extensively studied, little is known about *markdown* pricing under unknown demand. Recently, [4] and [9] independently showed that unimodality and Lipschitzness in *revenue* function (defined as the price times the mean demand) are necessary to achieve sublinear regret. In this setting, any reasonable policy must reduce the price at a moderate rate and stop only when there is sufficient evidence for *overshooting* the optimal price. By selecting suitable parameters, [4, 9] showed that the above policy is indeed optimal, achieving a tight $T^{3/4}$ regret bound.

However, in practice, it is usually assumed that the demand function has certain parametric forms, such as linear, exponential or logit form, allowing faster demand learning and suggesting lower regret bounds. This motivates our first question:

> Q1) Can we strengthen the $T^{3/4}$ regret bound for markdown pricing in this setting?

36th Conference on Neural Information Processing Systems (NeurIPS 2022).

Noticeably, the tight $T^{3/4}$ regret bound in [4, 9] this bound is asymptotically higher than $T^{2/3}$, the known regret bound for *unconstrained* pricing ([11]), highlighting the extra complexity caused by monotonicity constraint. Thus, we naturally arrive at our second question.

   Q2) Is markdown pricing still harder than unconstrained pricing under parametric assumptions?

Or, more precisely, can we still show a *separation* between markdown and unconstrained pricing, under various parametric assumptions? While one may answer these two questions for specific parametric families, it is tempting to find a unified framework that captures the hardness of demand learning for a given parametric family. This motivates the following question.

Q3) Can we find a general framework to unify the regret bounds for different *categories* of families, rather than specific results for specific families?

In this work, we propose such a framework by introducing a concept called *markdown dimension*. We provide efficient markdown policies for each markdown dimension, which we also show to be *best* possible, thereby completely settling the problem of markdown pricing under unknown demand.

## 1.1 Our Contributions.

In this work, we make the following contributions.

1. **New Complexity Measure for Demand Learning:** We introduce a new concept called *markdown dimension*, which captures the complexity of performing markdown pricing on a family, answering Q3). Within this framework, we provide a complete settlement of the problem as specified below.

2. **Markdown Policies with Theoretical Guarantees:** For each finite markdown dimension $d \geq 0$, we present a efficient markdown pricing policy. Our policies proceed in *phases*, in which the seller learns the demand by selecting prices at a suitable spacing to estimate the true parameter, and then makes conservative decisions. We show that for $d = 0$ and $d \geq 1$, our policies achieve regret $O(\log^2 T)$ and $\tilde{O}(T^{\frac{d}{d+1}})$ respectively, settling Q1).

3. **Tight Minimax Lower Bounds:** We complement our upper bounds with a matching lower bound for each markdown dimension $d$. More precisely, we show that $\Omega(\log^2 T)$ regret is tight for $d = 0$, which *separates* it from the known $O(\log T)$ regret bound without this monotonicity constraint (see [3]). For finite $d \geq 1$, we show an $\Omega(T^{d/(d+1)})$ lower bound, which not only matches our upper bound but is also asymptoticly higher than the tight $\tilde{\Theta}(T^{1/2})$ upper bound (see [3, 10]) without markdown constraint, settling Q2).

4. **Impact of Smoothness:** We go further to refine our bounds and investigate the impact of smoothness of reward function around the optimal price. We consider a *sensitivity* parameter $s$, which essentially says that an $\varepsilon$ distance away from the optimal price incurs $O(\varepsilon^s)$ regret, and can be verified to be at least 2 assuming the demand function is twice-differentiable. For both finite and infinite $d$, we extend our upper bounds to incorporate $s$. Moreover, for $d = \infty$, our tight $T^{\frac{2s+1}{3s+1}}$ regret bound is asymptoticly higher than that for unconstrained pricing, whose optimal regret is known to be $T^{\frac{s+1}{2s+1}}$ ([2]).

We highlight our results for $s = 2$ in red in Table 1, where $\Theta$ denotes tight upper and lower bounds.

| Markdown Dimension | Markdown Pricing | Unconstr. Pricing |
| --- | --- | --- |
| $d = 0$ | $\Theta(\log^2 T)$ | $\Theta(\log T)$ [3] |
| $1 \leq d < \infty$ | $\tilde{\Theta}(T^{d/(d+1)})$ | $\tilde{\Theta}(\sqrt{T})$ [3] |
| $d = \infty$ | $\tilde{\Theta}(T^{3/4})$ [4, 9] | $\tilde{\Theta}(T^{2/3})$ [11] |

Table 1: Regret bounds for $s = 2$.

## 1.2 Related Work

In the *Multi-Armed Bandit* (MAB) problem, the player is offered a finite set of arms, with each arm providing a random revenue from an unknown probability distribution specific to that arm. The

objective of the player is to maximize the total revenue earned by pulling a sequence of arms (e.g. [12]). Our pricing problem generalizes this framework by using an infinite action space $[0, 1]$ with each price $p$ corresponding to an action whose revenue is drawn from an unknown distribution with mean $R(p)$. In the *Lipschitz Bandit* problem (see, e.g., [1]), it is assumed that each $x \in [0, 1]$ corresponds to an arm with mean reward $\mu(x)$, and $\mu$ satisfies the Lipschitz condition, i.e. $|\mu(x) - \mu(y)| \leq L|x - y|$ for some constant $L > 0$. For for one-dimensional Lipschitz Bandits, there is a known $\tilde{\Theta}(T^{2/3})$ regret bound ([11]).

Recently there is an emerging line of work on bandits with monotonicity constraint. [4] and [9] recently independently considered the markdown pricing problem under unknown demand function and proved a tight $T^{3/4}$ regret bound under the minimal assumptions – Lipschitzness and unimodality on the revenue functions. Motivated by fairness constraints, [8] and [15] considered a general online convex optimization problem where the action sequence is required to be monotone.

Other requirements on the arm sequence motivated by practical problems are also considered in the literature. For example, [5, 14] and [16] are motivated by a similar concern in dynamic pricing: customers are hostile to frequent price changes. They propose algorithms that have a limited number of switches and study the impact on the regret. Motivated by Phase I clinical trials, [6] study the best arm identification problem when the reward of the arms are monotone and the goal is to identify the arm closest to a threshold.

## 2 Model and Assumptions

We begin by formally stating our model. In this work we assume an unlimited supply of a single product. Given a discrete time horizon of $T$ rounds, in each round $t$, the policy (representing the "seller") selects a price $p_t$ (the particular interval $[0, 1]$ is without loss of generality, by scaling). The demand $D_t$ in this round is then independently drawn from a fixed distribution with *unknown* mean $D(p_t)$, and the policy receives revenue (or *reward*, which we will use interchangeably) $p_t$ for each unit sold, and hence a total of $p_t \cdot D_t$ revenue in this round. The only constraint the policy must satisfy is the *markdown* constraint: $p_1 \geq \cdots \geq p_T$ with probability 1.

The function $D(p)$ which maps each price $p$ to the mean demand at this price is known as the *demand function*. For any policy[1] $\pi$ and demand function $D(\cdot)$, we use $r(\pi, D)$ to denote the expected total reward of $\pi$ under $D$. Rather than evaluating policies directly in terms of $r(\pi, D)$, it is more informative (and ubiquitous in the literature on MAB) to measure performance using the notion of *regret* with respect to a certain idealized benchmark. Specifically, since we assumed unlimited supply, when the true reward function is known, the seller simply always selects a revenue-maximizing price $p_D^* = \arg\max_{p \in [0,1]} p \cdot D(p)$ at each round, and we denote this maximal reward rate to be $r_D^* = \max_{p \in [0,1]} p \cdot D(p)$. The regret of a policy is then defined with respect to this quantity, and we seek to bound the *worst-case* value over a given family of demand functions.

**Definition 1** (Regret). For any policy $\pi$ and demand function $D$, define the *regret* of policy $\pi$ under $D$ to be $\text{Reg}(\pi, D) := r_D^* \cdot T - r(\pi, D)$. For any given family $\mathcal{F}$ of demand functions, the *worst-case regret* (or simply *regret*) of policy $\pi$ for family $\mathcal{F}$ is $\text{Reg}(\pi, \mathcal{F}) := \sup_{D \in \mathcal{F}} \text{Reg}(\pi, D)$.

### 2.1 Basic Assumptions

Now we state the common assumptions that all of our results rely on. A demand function $D(\cdot)$ is naturally associated with a *revenue function* (or, *reward function*) $R(p) = p \cdot D(p)$. Sometimes it will be convenient to work directly with the revenue functions, in which cases we write $r(\pi, R)$ as the regret of under reward function is $R$.

**Definition 2** (Optimal Price Mapping). Let $\mathcal{F}$ be a set of functions defined on some set $S \subseteq \mathbb{R}$. For any function $R : S \to \mathbb{R}$, let $M(R)$ be the subset of its global maxima on $S$. The *optimal price mapping* $p^* : \mathcal{F} \to S$ is defined as $p^*(R) = \inf M(R)$.

By elementary topology, if the domain $S$ is compact, then $M(R)$ is also compact, so the infimum of $M(R)$ can be attained, and hence $p^*(R)$ is also a global maximum of $R$. We first introduce a standard assumption (see e.g. [3]), which assumes the derivative of $R$ vanishes at $p^*(R)$.

---

[1]Formally, a *policy* is a sequence $\pi = (\pi_t)_{t \in [T]}$ of mappings where $\pi_t : [0, 1]^{t-1} \times \mathbb{R}_{>0}^{t-1} \to [0, 1]$ corresponds to the decision made at time $t$, based on the realized demands and prices selected up till time $t - 1$.

**Assumption 1** (Vanishing Derivative). We assume that every reward function $R$ is differentiable on its domain, and moreover, $R'(p^*(R)) = 0$.

Under Assumption 1, if, in addition, $R$ is twice differentiable, then by Taylor expansion around $p^*$,

$$|R(p^*) - R(p^* + \varepsilon)| = \frac{1}{2} R''(p^*) \cdot \varepsilon^2 + o(\varepsilon^2),$$

for small $\varepsilon$. Thus, if a policy overshoots the optimal price by $\varepsilon$, then only an $O(\varepsilon^2)$ loss is incurred in each round. We next introduce a *distributional* assumption.

**Definition 3** (Subgaussian Random Variable). The *subgaussian norm* of a random variable $X$ is

$$\|X\|_{\psi_2} := \inf\{c > 0 : \mathbb{E}[e^{X^2/c^2}] \leq 2\},$$

and $X$ is said to be *subgaussian* if $\|X\|_{\psi_2} < \infty$.

**Assumption 2** (Subgaussian noise). There exists a constant $C_{sg} > 0$ such that under any true demand function and any price $p$, the random demand $X$ at price $p$ satisfies $\|X\|_{\psi_2} \leq C_{sg}$.

## 2.2 Warm-up: Markdown Pricing on Linear Demand

Before introducing our concept of markdown dimension, we first consider linear demand as a warm up. Let $D(x; \theta) = \theta_1 - \theta_2 x$ for $x \in [\frac{1}{2}, 1]$. Consider the following natural markdown policy for $\mathcal{F} = \{D(x; \theta) : \theta_1, \theta_2 \in [\frac{1}{2}, 1]\}$. Choose two *sample prices* $p_1, p_2$ close to 1 and collect sufficiently many samples at each, say with empirical mean demands $\bar{d}_1, \bar{d}_2$. Then, compute the unique $\hat{\theta}$ satisfying $D(p_i; \hat{\theta}) = \bar{d}_i$ for $i = 1, 2$. Finally, select the optimal price of $D(\cdot; \hat{\theta})$ in remaining rounds.

This policy is "robust" in the following sense. Suppose $\bar{d}_i$ deviate from $d_i := D(p_i)$ by $\sim \delta$, then the estimation error in $(a, b)$ is $\sim \frac{\delta}{|p_1 - p_2|}$, i.e., proportional to $\delta$ and inverse proportional to the gap between the two sample prices.

Our markdown dimension generalizes the above idea. Informally, a family, say parametrized by $\theta$, has markdown dimension $d$ if the estimation of $\theta$ errs by $O\left(\frac{\delta}{h^d}\right)$ on a set of sample prices spaced at distance $h$ apart. We next introduce the formal definition.

## 2.3 Identifiability and Robust Parametrization

Intuitively, the exploration-exploitation trade-off for markdown pricing becomes harder to manage as the given family becomes more complex. Consider, for example, linear demand functions. If each function takes the form $D(p; c) = 1 - cp$ where only $c \in (0, 1)$ is unknown and $p \in [0, 1]$, then the seller simply needs to estimate the (negative) slope $c$ by sampling sufficiently many times at $p = 1$.

In contrast, if each function takes the form $D(p; a, b) = a - bp$ where *both* parameters $a, b$ are unknown, then sampling at one price would *not* suffice. Rather, one needs to select (at least) two distinct prices to estimate $a, b$, thereby facing the following dilemma. Suppose the two prices $p < p'$ selected are far apart. Then, $p$ may be far away from the optimal price $p^*$ since $p^*$ may be close to $p'$, resulting in high regret. Otherwise, when those prices are close by, the demand learning requires a high volume of samples, which potentially also leads to a high regret.

Thus we reach a natural question: can we introduce a complexity index to measure the difficulty of performing markdown pricing on a given family, and then provide tight regret bounds in terms of this complexity index? In this work, we propose a complexity index, called *markdown dimension*, and provide nearly-optimal regret bounds in terms of the markdown dimension of the given family of demand functions. The formal definition relies on other two concepts, the *identifiability* of a family, and the *robustness* of a parametrization, which we introduce in the next two subsections.

**Identifiability** Our notion of identifiability generalizes a key property of single-variable polynomials, that every degree-$d$ polynomial can be uniquely determined by its values at *any* $(d + 1)$ points. To present the formal definition, we first introduce a mapping which, for a fixed subset of prices, assigns each demand function a *profile* based on its values at those prices.

**Definition 4** (Profile Mapping). Consider a set $\mathcal{F}$ of real-valued functions defined on $A \subseteq \mathbb{R}$. For any fixed $\mathbf{p} = (p_0, p_1, ..., p_d) \in A^{d+1}$, the *profile-mapping* with respect to $\mathbf{p}$ is defined as

$$\Phi_{\mathbf{p}} : \mathcal{F} \to \mathbb{R}^d,$$
$$D \mapsto (D(p_0), D(p_1), ..., D(p_d)).$$

We may subsequently call $\Phi_{\mathbf{p}}(D)$ the *profile* of function $D$ with respect to $\mathbf{p}$. In words, the above says that every function in $\mathcal{F}$ be assigned a unique profile at any $(d+1)$ distinct points ("prices").

**Definition 5** (Identifiability). The family $\mathcal{F}$ is *d-identifiable*, if for any *distinct* $p_0, p_1..., p_d \in S$, the profile mapping $\Phi_{p_0, ..., p_d}$ is injective, i.e. distinct functions in $\mathcal{F}$ are mapped to distinct profiles.

In particular, if a family is $d$-identifiable, then for any distinct $p_0, p_1..., p_d$ the inverse profile-mapping $\Phi_{\mathbf{p}}^{-1} : \mathcal{R}_p \to \mathcal{F}$ exists, where $\mathcal{R}_p$ is the range of the mapping $\Phi_{\mathbf{p}}$.

**Robust Parametrization** We first formally define a *parametrization*.

**Definition 6** (Parametrization). An *order-m parametrization* for a family $\mathcal{F}$ of functions is any one-to-one mapping from a compact set $\Theta \subseteq \mathbb{R}^m$ to $\mathcal{F}$. Moreover, each $\theta \in \Theta$ is called a *parameter*.

By abuse of notations, we may use $D(p; \theta)$ to denote the function $D(p)$ that parameter $\theta$ corresponds to. As a standard assumption (see e.g. [3]), we also assume that the parameter set $\Theta$ to be compact, which leads to many favorable properties.

**Assumption 3** (Compact Domain). The domain $\Theta$ of the parametrization is compact.

Under this assumption, the demand functions in $\mathcal{F}$ are bounded, and thus we may without loss of generality also scale the range (i.e. target space) of those functions to be $[0, 1]$.

**Assumption 4** (Smoothness). The mapping $D : [0, 1] \times \Theta \to \mathbb{R}$ is twice-differentiable and admits continuous second partial derivative. In particular, since $[0, 1] \times \Theta$ is compact, under the above assumption, there exist constants $C^{(j)} > 0$ such that the $j$-th derivative satisfies $|D^{(j)}(p, \theta)| \leq C^{(j)}$ for any $(p, \theta) \in [0, 1] \times \Theta$ and $j = 0, 1, 2$.

Recall that we previously defined the optimal price mapping $p^*$ from $\mathcal{F}$ to the domain, $[0, 1]$, of the reward functions. Now that we introduced a parametrization, by abuse of notation we may view the mapping $p^*$ as being defined on $\Theta \subseteq \mathbb{R}^m$. Formally, let $M(\theta)$ be the set of global maxima of $R(x; \theta)$, then $p^*(\theta) = \inf R(x; \theta)$. The following assumption allows us to "propagate" the error of parameter estimation to optimal price estimation.

**Assumption 5** (Lipschitz Optimal Price Mapping). The optimal price mapping $p^* : \Theta \to [0, 1]$ is $C^*$-Lipschitz for some constant $C^* > 0$.

This assumption has appeared in the previous literature on parametric demand learning, see e.g. Assumption 1(c) of [3]. Moreover, it is satisfied by many commonly used demand functions such as linear, exponential and logit demand. For instance, let $D(p; c) = 1 - cp$ where $p, c \in [0, 1]$, then $p^*(c) = \min\{\frac{1}{2c}, 1\}$, which is 1-Lipschitz. Nonetheless, this assumption is somewhat unnatural as it is not made on the parametrization directly. We leave it open whether it can be made more directly.

The final ingredient for robust parametrization is motivated by the following robustness of the *natural parametrization* $D(p; \theta) = \sum_{j=0}^{d} \theta_j p^j$ for polynomials. Consider any distinct prices $p_0, p_1, ..., p_d$, and any $d + 1$ real numbers $y_0, y_1, ..., y_d$ representing, for example, the mean reward at each $p_i$. We may then uniquely determine a degree-$d$ polynomial by solving the linear equation

$$V_{\mathbf{p}} \theta = \mathbf{y}$$

where $\theta = (\theta_0, \theta_1 ..., \theta_d)^T$, $\mathbf{y} = (y_0, y_1, ..., y_d)^T$ and $V_{\mathbf{p}} := V(p_0, ..., p_d)$ is the *Vandermonde* matrix. One can easily verify that when $p_i$'s are distinct, $V_{\mathbf{p}}$ is invertible, and hence $\theta = V_{\mathbf{p}}^{-1} \mathbf{y}$. Next consider the effect of a perturbation on $\mathbf{y}$, in terms of the following *separability* parameter.

**Definition 7** (Separability). For any $\mathbf{p} = (p_0, ..., p_d) \in \mathbb{R}^{d+1}$, define $h(\mathbf{p}) := \min_{i \neq j} |p_i - p_j|$.

To motivate the notion of robust parametrization, first consider a result specific to polynomial demand.

**Proposition 1** (Robustness of Polynomial Parametrization). There exist constants $C_1, C_2 > 0$ such that for any $\mathbf{p} \in [0,1]^{d+1}$ with $0 < h(\mathbf{p}) \le C_1$, any $\varepsilon \le C_1$, and $\mathbf{y}, \hat{\mathbf{y}} \in \mathcal{R}_p$ with $\|\mathbf{y} - \hat{\mathbf{y}}\|_\infty \le C_1$, it holds that

$$\|V_{\mathbf{p}}^{-1} \mathbf{y} - V_{\mathbf{p}}^{-1} \hat{\mathbf{y}}\|_\infty \le C_2 \cdot \|\mathbf{y} - \hat{\mathbf{y}}\|_\infty \cdot h(\mathbf{p})^{-d}. \tag{1}$$

More concretely, let $D(p; \theta)$ be the underlying polynomial demand function, then $\mathbf{y} = V_{\mathbf{p}} \cdot \theta$ are the mean demands at the prices in $\mathbf{p}$. Suppose a learner observes empirical mean demands $\hat{\mathbf{y}}$ at $\mathbf{p}$, then a natural estimator for $\theta$ is simply $V_{\mathbf{p}}^{-1} \cdot \hat{\mathbf{y}}$. Our Proposition 1 can then be viewed as an upper bound on the error of this estimator, in terms of $h(\mathbf{p})$ and $\mathbf{y} - \hat{\mathbf{y}}$.

In order to achieve sublinear regret, the value $h = h(\mathbf{p})$ must be chosen to be $o(1)$ as $T$ goes to infinity. Thus, the dependence on $h$ crucially affects our regret bounds. Proposition 1 establishes a nice property for polynomials, that the estimation error increases in the order of $(\frac{1}{h})^d$, as $h \to 0^+$. We introduce *robust parametrization* by generalizing this property beyond polynomials. Loosely, an order-$d$ parametrization is *robust*, if it admits a similar error bound to (1).

**Definition 8** (Robust Parametrization). An order-$d$ parametrization $\theta : \Theta \to \mathcal{F}$ is *robust*, if
(1) it satisfies Assumptions 3, 4 and 5, and
(2) there exist constants $C_1, C_2 > 0$ such that for any $\mathbf{p} \in \mathbb{R}^{d+1}$ with $0 < h(\mathbf{p}) \le C_1$ and any $y, y' \in \mathcal{R}_p$ with $\|\mathbf{y} - \mathbf{y}'\|_\infty \le C_1$, it holds that

$$\|\Phi_{\mathbf{p}}^{-1}(\mathbf{y}) - \Phi_{\mathbf{p}}^{-1}(\mathbf{y}')\|_\infty \le C_2 \cdot \|\mathbf{y} - \mathbf{y}'\|_\infty \cdot h(\mathbf{p})^{-d}. \tag{2}$$

In particular, when $d = 0$, inequality (2) simply says $\Phi_p^{-1}$ is $C_2$-Lipschitz.

## 2.4 Markdown Dimension

Now we are ready to define the markdown dimension.

**Definition 9** (Markdown Dimension). The *markdown dimension* (or simply *dimension*) for a family $\mathcal{F}$ of functions, denoted $d(\mathcal{F})$, is the minimum integer $d \ge 0$ such that $\mathcal{F}$ is (i) $d$-identifiable, and (ii) admits a robust order-$d$ parametrization. If no finite $d$ satisfies the above conditions, then $d(\mathcal{F}) = \infty$.

We further illustrate our definition by considering the dimensions of some commonly used families. As the simplest family, one may verify that our definition of 0-dimensional family (under our assumptions) is equivalent to the *separable* family as defined in Section 4 of [3]. We provide more concrete examples below, whose proof can found in Appendix A.

**Proposition 2.** Suppose $D(x; \theta) = \sum_{j=0}^{d} \theta_j x^j$ and $\mathcal{F} = \{D(x; \theta) : \theta \in \Theta\}$ satisfies Assumptions 3, 4 and 5. Then, $d(\mathcal{F}) = d$.

**Proposition 3** (0-Dimensional Families). The following families $\mathcal{F}_1, \mathcal{F}_2$ and $\mathcal{F}_3$ are 0-dimensional.
(1) **[Single-Parameter Linear Demand]** Let $D_a(x) = 1 - ax$ for $x \in \left[\frac{1}{2}, 1\right]$ and $\mathcal{F}_1 = \{D_a(x) : a \in \left[\frac{1}{2}, 1\right]\}$.
(2) **[Exponential Demand]** Let $D_a(x) = e^{1-ax}$ for $x \in \left[\frac{1}{2}, 1\right]$ and $\mathcal{F}_2 = \{D_a(x) : a \in \left[\frac{1}{2}, \frac{3}{4}\right]\}$.
(3) **[Logit Demand]** Let $D_a(x) = \frac{e^{1-ax}}{1+e^{1-ax}}$ for $x \in \left[\frac{1}{2}, 1\right]$ and $\mathcal{F}_3 = \{D_a(x) : a \in \left[\frac{1}{2}, 1\right]\}$.

Finally, we observe that by our definition, if a family of functions is not $d$-identifiable for any $d$. For example, for any $d$ distinct prices, there exists multiple (more precisely, infinitely many) Lipschitz functions having the same values on these $d$ prices.

**Proposition 4.** Let $\mathcal{F}$ be the set of all 1-Lipschitz functions on $[0,1]$, then $d(\mathcal{F}) = \infty$.

## 2.5 Sensitivity

Consider the Taylor expansion of a reward function $R(x)$ around an optimal price $p^*$:

$$R(p) = R(p^*) + 0 + \frac{1}{2!} R''(p^*)(p - p^*)^2 + \frac{1}{3!} R^{(3)}(p^*)(p - p^*)^3 + ...$$

Suppose the first nonzero derivative is $R^{(k)}(p^*)$. Then, the higher $k$, the less the revenue is *sensitive* to overshooting (i.e. $p < p^*$). Our notion of *sensitivity* measures how fast the revenue function changes at the optimum.

**Definition 10** (Sensitivity). A reward function $R$ is *s-sensitive* if it is $(s+1)$-differentiable with $R^{(1)}(p^*(R)) = ... = R^{(s-1)}(p^*(R)) = 0$ and $R^{(s)}(p^*(R)) < 0$. A family $\mathcal{F}$ of reward functions is called *s-sensitive* if

(a) every $R \in \mathcal{F}$ is $s$-sensitive,

(b) it admits a parametrization $R(x; \theta)$ satisfying Assumptions 3 to 5, and

(c) there is a constant $C_6 > 0$ such that $R^{(s)}(p^*(R)) \leq -C_6 < 0$ for any $R \in \mathcal{F}$,

(d) for $j = 0, 1, ..., s$, there exists a constant $C^j$ s.t. $|R^{(j)}(x; \theta)| \leq C^j$ for all $x \in [0, 1]$ and $\theta \in \Theta$.

For example, let $R(x; \theta) = \theta - |\frac{1}{2} - x|^s$ for $x \in [0, 1]$ and any $\theta \in \mathbb{R}$, then $\{R(x; \theta) : \theta \in [\frac{1}{2}, 1]\}$ is an $s$-sensitive family. Note that by Taylor's Theorem, $|R(p^* + \varepsilon) - R(p^*)| \leq \frac{C_s}{s!}|\varepsilon|^s$. Consequently, if a policy overshoots or undershoots the optimal price by $\varepsilon$, the regret *per round* is only $O(\varepsilon^s)$, which is asymptoticly lower than the per-round regret $O(\varepsilon^2)$ without the sensitivity assumption. We address a natural question: how does the regret bounds for markdown pricing change as $s$ increases?

## 3 Zero-Dimensional Family

We start with the simplest case, $0$-dimensional demand functions. As opposed to the *optimism* in the face of uncertainty in UCB policies, our *Cautious Myopic* policy adopts *conservatism* in the face of uncertainty. More precisely, we partition the time horizon so that the *phase $j$* consists of $t_j := \lceil 9^j \log T \rceil$ rounds, and thus in total there are $K = O(\log T - \log \log T)$ phases. In each phase, the policy estimates the true parameter $\theta^*$ using the observations from the last phase, and builds a confidence interval around $\theta^*$, which depends on the number of length of this phase and also the constant $C_{sg}$ as defined in Assumption 2. Then, in the next phase, the policy selects the *largest* optimal price of any parameter $\theta$ in the confidence interval. We write $t^{(j)} := \sum_{k=0}^{j} t_k$ and for convenience $t^{(0)} = 0$, and formally state this policy in Algorithm 1.

---

**Algorithm 1** Cautious Myopic (CM) Policy.

---

1: Input: a family $\mathcal{F}$ of demand functions and time horizon $T$.

2: $p_1 \leftarrow 1$          % Initialization

3: **for** $j = 1, ..., K$ **do**

4:     **for** $t = t^{(j-1)} + 1, ..., t^{(j-1)} + t_j$ **do**

5:        $x_t \leftarrow p_j$        % Select $p_j$ for $t_j$ times in phase $j$

6:        Observe realized demand $D_t$

7:     **end for**

8:     $\bar{d}_j = \frac{1}{t_j} \sum_{\tau=1}^{t_j} D_{t^{(j-1)}+\tau}$        % Empirical mean demand in phase $j$

9:     $\hat{\theta}_j \leftarrow \Phi_{p_j}^{-1}(\bar{d}_j)$        % Estimate parameter

10:     $w_j \leftarrow 4C_2 \cdot C_{sg}\sqrt{\frac{\log T}{t_j}}$        % Width of the confidence interval

11:     $\tilde{p}_{j+1} \leftarrow \max\{p^*(\theta) : |\theta - \hat{\theta}_j| \leq w_j\}\}$        % Conservative estimation of the optimal price

12:     $p_{j+1} \leftarrow \min\{\tilde{p}_{j+1}, p_j\}$        % Ensure monotonicity

13: **end for**

---

**Theorem 1** (Zero-dimensional Upper Bound). Let $\mathcal{F}$ be any $0$-dimensional, $s$-sensitive family of demand functions. Then the Cautious Myopic (CM) Policy has regret

$$\text{Reg}(\text{CM}, \mathcal{F}) = \begin{cases} O(\log^2 T), & \text{if } s = 2, \\ O(\log T), & \text{if } s > 2. \end{cases}$$

It is worth noting that this bound is asymptotically higher than the $O(\log T)$ upper bound for unconstrained pricing ([3]). Intuitively, this is because the CM policy purposely keeps a distance from the estimated optimal price. But can we achieve $o(\log^2 T)$ regret by taking more risk or being more conservative? We answer this question negatively by showing that CM is indeed optimal, providing a *separation* unconstrainted and markdown pricing for $0$-dimensional demand families.

**Theorem 2** (Zero-Dimensional Lower Bound). There exists a $0$-dimensional $2$-sensitive family $\mathcal{F}$ such that for any policy $\pi$, we have $\text{Reg}(\pi, \mathcal{F}) = \Omega(\log^2 T)$.

# 4 Finite-Dimensional Family

For dimension $d$, the learner needs $d + 1$ distinct *sample prices*, as opposed to just one price when $d = 0$. This, however, incurs extra regret, since the optimal price may lie *between* these sample prices. Intuitively, a reasonable policy trades off between the overshooting risk and the learning rate. If the gap is large, the policy may learn the parameter efficiently, but there is potentially a higher regret due to overshooting, in case the true optimal price lies between the sample prices. On the other extreme, if the gap is small, there is less risk of overshooting but a slower rate of learning.

We introduce our Iterative Cautious Myopic (ICM) Policy (Algorithm 2) that strikes such balance nearly optimally, as we will soon see. The policy consists of $m$ phases. In phase $j \in [m]$, the policy selects $d$ *sample prices* evenly spaced at distance $h$, each for $T_j$ times. It then computes an estimated the optimal price $\hat{p}_j$ along with a confidence interval $[L_j, U_j]$, based on the observed demands.

To determine the initial price $p_{j+1}$ in the next phase, the policy considers three cases. Recall that the last price that the policy selects in phase $j$ is $p_j - dh$. We say a *good* event occurs, if $p_j - dh > U_j$, in which case we set $p_{j+1} = U_j$. In the *dangerous* event, the current price $p_j - dh$ is within the confidence interval, and we may have already overshot the optimal price. Thus, the policy needs to behave conservatively to prevent from overshooting further, so it selects $p_{j+1} = p_j - dh$. Finally in the *overshooting* event, the current price is already lower than $L_j$, and hence with high probability the policy has overshot. In this case, it immediately exits the exploration phase (i.e. the outer for-loop) and enters the exploitation phase, wherein it selects the current price in all remaining rounds.

---

**Algorithm 2** Iterative Cautious Myopic (ICM) Policy.

---

1: Input: $\mathcal{F}, m, \{T_j\}_{j \in [m]}, T$
2: $p_1 \leftarrow 1, L_0 \leftarrow 0, U_0 \leftarrow 1$ Initialization
3: **for** $j = 1, 2, ...m$ **do**
4:     **for** $k = 0, 1, ..., d$ **do**
5:         Select price $p_j - kh$ for $T_j$ times and observe $D_1, ..., D_{T_j}$
6:         $\bar{D}_k \leftarrow \frac{1}{T_j} \sum_{i=1}^{T_j} D_i$                % Mean demand at $p_j - kh$
7:     **end for**
8:     $\hat{\theta} \leftarrow \Phi_{p_j, ..., p_j - dh}^{-1}(\bar{D}_0, ..., \bar{D}_d)$         % Estimate Parameter
9:     $w_j \leftarrow 2h^{-d} \cdot C_2 \cdot C_{sg} \sqrt{\frac{d \log T}{T_j}}$     % Width of confidence interval
10:    $L_j \leftarrow \min\{p^*(\theta) : \|\theta - p^*(\hat{\theta})\|_2 \leq w_j\}$     % Lower confidence bound
11:    $U_j \leftarrow \max\{p^*(\theta) : \|\theta - p^*(\hat{\theta})\|_2 \leq w_j\}$     % Upper confidence bound
12:    **if** $U_j \leq p_j - dh$ **then**
13:       $p_{j+1} \leftarrow U_j$                  % Good event
14:    **end if**
15:    **if** $U_j > p_j - dh \geq L_j$ **then**
16:       $p_{j+1} \leftarrow p_j - dh$
17:    **end if**
18:    **if** $p_j - dh < L_j$ **then**
19:       Break                   % Overshooting event
20:    **end if**
21: **end for**
22: Select the current price in every future roundExploitation

---

**Theorem 3** (Upper Bound for Finite $d \geq 1$). Suppose $s = 2$, then for $h = T^{\frac{m}{m(d+1)+1}}$, $T_j = T^{\frac{md+j}{m(d+1)+1}}$ and $m = \log T$, we have $\text{Reg}(ICM, \mathcal{F}) = \tilde{O}\left(T^{\frac{d}{d+1}}\right)$. More generally, for any $s \geq 2$ and $m = \tilde{O}(1)$, there exists $T_1 < ... < T_m$ such that $\text{Reg}(ICM, \mathcal{F}) = \tilde{O}\left(T^{\rho(m,s,d)}\right)$ where

$$\rho(m, s, d) = \frac{1 + \left(1 + \frac{s}{2} + ... + \left(\frac{s}{2}\right)^{m-1}\right) d}{\left(\frac{s}{2}\right)^m + \left(1 + \frac{s}{2} + ... + \left(\frac{s}{2}\right)^{m-1}\right) \cdot (d+1)}.$$

We complement the upper bound with an nearly tight lower bound for $s = 2$, stated below.

**Theorem 4** (Lower Bound for Finite $d \geq 1$). For any $d \geq 2$, there exists a $d$-dimensional family $\mathcal{F}$ of demand functions on $[0, 1]$ such that for any markdown policy $\pi$, we have $\text{Reg}(\pi, \mathcal{F}) = \Omega(T^{\frac{d}{d+1}})$.

In our proof, for each $d \geq 1$ we construct a sub-family of $(d+1)$-degree decreasing polynomial demand functions – which is also $d$-dimensional – on which any policy suffers regret $\Omega(T^{\frac{d}{d+1}})$.

At first glance, when $d \geq 4$ this lower bound seems to contradict the $T^{3/4}$ upper bound for unimodal, Lipschitz reward family ([4, 9]). This is because the unimodality assumption no longer holds. In fact, in general a degree-$d$ polynomial with $d \geq 4$ is $d$-dimensional but in general not unimodal. Another interesting case is $d = 1$. In this case, a degree-$d$ polynomial (i.e. linear) demand corresponds to a unimodal (in fact, quadratic) reward function, hence the $T^{3/4}$ upper bound from [4, 9] applies. Theorem 3 shows that this bound can be improved to $T^{1/2}$.

We conclude the section by pointing out a limitation of our results. For each $d \geq 1$, our regret bounds is tight in the *exponent* of $T$, however, the constants in the big-O have different dependence on $d$. In fact, the upper and lower bound become exponentially higher and lower in $d$ respectively as $d$ grows.

## 5 Infinite Dimensional Family

When $d = \infty$, it is more convenient to work with the reward function $R(x) := x \cdot D(x)$ instead of the demand function. In contrast to *unconstrained* pricing, there is no *markdown* policy that achieves $o(T)$ regret on the family of Lipschitz reward functions ([4, 9]), since the reward function may have multiple local maxima. Nonetheless, [4] and [9] showed that if in addition we assume the reward function to be unimodal (which is satisfied by many commonly used families), then a tight $\tilde{\Theta}(T^{3/4})$ regret is achievable. In this setting, the problem boils down to finding the unique local optimum $p^*$ of the true revenue function. Specifically, the lower bound is established by considering reward functions where the reward rate may change *abruptly* at $p^*$. We next show that the regret bound can be improved if we assume that the reward functions change smoothly.

---

**Algorithm 3** Uniform Elimination Policy ($\text{UE}_{m,\Delta}$).

1: Input: $T, \Delta, m > 0$
2: Initialize: $L_{\max} \leftarrow 0$, $w \leftarrow 2C_{sg}\sqrt{\frac{\log T}{m}}$
3: **for** $j = 0, 1, 2, ..., \lceil \Delta^{-1} \rceil$ **do**
4:     $x_j \leftarrow 1 - j\Delta$
5:     Select price $x_j$ for the next $m$ rounds and observe rewards $Z_1^j, ..., Z_m^j$
6:     $\bar{\mu}_j \leftarrow \frac{1}{m}\sum_{i=1}^m Z_i^j$            % Compute mean rewards
7:     $[L_j, U_j] \leftarrow [\bar{\mu}_j - w, \bar{\mu} + w]$ Compute confidence interval for reward at current price
8:     **if** $L_j > L_{\max}$ **then**
9:         $L_{\max} \leftarrow L_j$           % Keep track of the highest $L_j$
10:     **end if**
11:     **if** $U_j < L_{\max}$Exploration phase ends **then**
12:         $h \leftarrow j$           % Define the *halting price*
13:         Break
14:     **end if**
15: **end for**
16: Select price $x_h$ in all remaining rounds.         % Exploitation phase

---

**Theorem 5** (Upper Bound for Infinite-Dimensional Family)**.** Let $\mathcal{F}_s^U$ be the family of unimodal and $s$-sensitive reward functions. For any $s \geq 2$, there exists suitable choices of $m$ and $\Delta$ such that $\text{Reg}(\text{UE}_{m,\Delta}, \mathcal{F}_s^U) = O(T^{\frac{2s+1}{3s+1}})$.

We complement the above theorem with a lower bound in terms of both $s$ and $T$, that matches the upper bound in Theorem 5 for every $s \geq 2$.

**Theorem 6** (Lower Bound for $s$-Sensitive Family)**.** For any $s \geq 2$, there is a family $\mathcal{F}$ of $s$-sensitive unimodal revenue curves such that any markdown policy $\pi$ satisfies $\text{Reg}(\pi, \mathcal{F}) = \Omega(T^{\frac{2s+1}{3s+1}})$.

As a final remark, we observe that all our results assume that $d$ is given. If instead the learner only has partial information, for example an upper bound $d_{\max}$ on $d$, then she may simply choose the policy for $d_{\max}$ and obtain the corresponding guarantee $d_{\max}$.

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

# A   Omitted Proofs in Section 2

## A.1   Proof of Propositions 1 and 2

The infinite norm of a matrix is simply the maximum absolute row sum of the matrix.

**Definition 11** (Matrix $L_\infty$-Norm)**.** For any $A \in \mathbb{R}^{n \times n}$, define $\|A\|_\infty = \max_{i \in [n]} \sum_{j \in [n]} |a_{ij}|$.

We will need the following forklore inequality (see e.g. [17]).

**Lemma 1.** For any $A \in \mathbb{R}^{n \times n}$ and $v \in \mathbb{R}^n$, it holds $\|Av\|_\infty \le \|A\|_\infty \cdot \|v\|_\infty$.

The inverse of Vandermonde matrix admits a (somewhat complicated) closed-form expression, which leads to the following upper bound on its norm.

**Theorem 7** (Gautschi [7])**.** Let $x_1, .., x_d$ be distinct real numbers and $V = V(x_1, ..., x_d)$ be the corresponding Vandermonde matrix, then

$$\|V^{-1}\|_\infty \le \max_{i \in [n]} \prod_{j \neq i} \frac{1 + |x_j|}{|x_j - x_i|}.$$

Recall that for any $\mathbf{p} = (p_0, p_1, ..., p_d)$, we have defined $h(\mathbf{p}) = \min_{i \neq j}\{|p_i - p_j|\}$. From Theorem 7, we immediately obtain the following bound on $\|V_{\mathbf{p}}^{-1}\|_\infty$ in terms of $h(\mathbf{p})$.

**Corollary 1.** For any $\mathbf{p} = (p_0, p_1, ..., p_d) \in [0, 1]^{d+1}$ with distinct entries, it holds that

$$\|V_{\mathbf{p}}^{-1}\|_\infty \le \frac{2^d}{h(\mathbf{p})^d}.$$

*Proof.* Fix any $i \in [n]$. Since $0 \le p_j \le 1$ for any $j \in [n]$, we have $1 + |x_j| \le 2$, and thus

$$\prod_{j=1, j \neq i} \frac{1 + |x_j|}{|x_j - x_i|} \le \frac{2^d}{h(\mathbf{p})^d},$$

and the proof completes by noticing that $i$ is arbitrary. $\square$

**Proof of Proposition 1 and 2.** For any $\mathbf{p} = (p_0, p_1, ..., p_d) \in [0, 1]^{d+1}$ with distinct entries. Let $y, y' \in \mathcal{R}(V_{\mathbf{p}})$, then

$$\|V_{\mathbf{p}}^{-1} y' - V_{\mathbf{p}}^{-1} y\|_\infty \le \|V_{\mathbf{p}}^{-1}\|_\infty \cdot \|y' - y\|_\infty \le \frac{2^d}{h(\mathbf{p})^d} \cdot \|y' - y\|_\infty,$$

where the first inequality follows from by Lemma 1 and the second follows from Corollary 1. The proof completes by selecting the constant $C_2$ to be $2^d$, and Proposition 1 follows. Proposition 2 also follows immediately since we have just shown that the natural parametrization of polynomial is indeed robust. $\square$

## A.2   Proof of Proposition 3 (a)

We first observe that $\mathcal{F}_1$ is 0-identifiable, i.e., for any fixed $x \in [\frac{1}{2}, 1]$, $D(x; a)$ is injective in $a$. In fact, for any $a, a' \in [\frac{1}{2}, 1]$, $D(x; a) = D(x; a')$ amounts to $x \cdot (a - a') = 0$, which implies $a = a'$ since $x \neq 0$.

It remains to show that the order-0 parametrization $a \mapsto D(\cdot; a)$ is robust. This splits into the following steps, as required in Definition 8:

(1) **[Lipschitz Optimal Price Mapping]** By simple calculation, one can verify that for any $a \in [\frac{1}{2}, 1]$, the reward function $R(x; a) = a \cdot (1 - ax)$ admits a unique optimal price $p^*(a) = \frac{1}{2a}$, which is 2-Lipschitz.

(2) **[Robust Under Perturbation]** For any fixed price $x \in [\frac{1}{2}, 1]$, by definition of the profile mapping $\Phi$, we have $\Phi_x(a) = 1 - ax$. For any $y \in \mathcal{R}(\Phi_x)$, where we recall that $\mathcal{R}(\cdot)$ denotes the range of a mapping, we have $\Phi_x^{-1}(y) = \frac{1-y}{x}$. Thus, for any $y, y' \in \mathcal{R}(\Phi_x)$,

$$\left| \Phi_x^{-1}(y) - \Phi_x^{-1}(y') \right| = \left| \frac{1-y}{x} - \frac{1-y'}{x} \right| = \frac{|y - y'|}{x} \le 2 \cdot |y - y'|.$$

Therefore, the order-0 parametrization $a \mapsto D(\cdot; a)$ is robust, and thus $\mathcal{F}_1$ is 0-dimensional. $\square$

## A.3 Proof of Proposition 3 (b)

Write $D(x; a) = e^{1-ax}$. The following will be useful for (b) and (c), whose proof follows directly from the definition of identifiability.

**Lemma 2.** Let $\{D(x; \theta) : \theta \in \Theta\}$ be a 0-identifiable family, and $g : \mathbb{R} \to \mathbb{R}$ be strictly monotone, then $\{g(D(x; \theta)) : \theta \in \Theta\}$ is also 0-identifiable.

Recalling that $\mathcal{F}_1$ is 0-identifiable, and $g(z) = e^z$ is strictly increasing, by Lemma 2 we deduce that $\mathcal{F}_2$ is also 0-identifiable. It remains to show that $a \mapsto D(\cdot; a)$ is robust for $\mathcal{F}_2$. This splits into the following steps, as required in Definition 8:

(1) **[Lipschitz Optimal Price Mapping]** For any $a$, the corresponding reward function is $R(x; a) = a \cdot e^{1-ax}$. A price $x$ is then local optimum iff the derivative of $R'(x; a) = (1 - ax)e^{1-ax} = 0$, i.e., $x = \frac{1}{2a}$, which has just been verified to be Lipschitz in the proof for (a).

(2) **[Robust Under Perturbation]** For any fixed price $x \in [\frac{1}{2}, 1]$, we have $\Phi_x(a) = e^{1-ax}$, and therefore for any $y \in \mathcal{R}(\Phi_x)$, we have $\Phi_x^{-1}(y) = \frac{1-\ln y}{x}$. Thus, for any $y, y' \in \mathcal{R}(\Phi_x)$,

$$\left| \Phi_x^{-1}(y) - \Phi_x^{-1}(y') \right| = \left| \frac{1 - \ln y}{x} - \frac{1 - \ln y'}{x} \right| = \frac{|\ln y - \ln y'|}{x}. \tag{3}$$

Finally, to bound the above, note that $e^{\frac{1}{4}} \leq e^{1-ax} \leq e^{\frac{3}{4}}$ for any $x \in [\frac{1}{2}, 1]$ and $a \in [\frac{1}{2}, \frac{3}{4}]$, i.e., $\mathcal{R}(\Phi_x) \subseteq \left[ e^{\frac{1}{4}}, e^{\frac{3}{4}} \right]$. Thus, for any $x \in [\frac{1}{2}, 1]$,

$$(3) \leq \frac{\max_{t \in \mathcal{R}(\Phi_x)} \ln' t}{x} \cdot |y - y'| = 2e^{-\frac{1}{4}} \cdot |y - y'|.$$

Therefore the order-0 parametrization $a \mapsto D(\cdot; a)$ is robust, and hence $\mathcal{F}_2$ is 0-dimensional. $\square$

## A.4 Proof of Proposition 3 (c)

The 0-identifiability again follows from Lemma 2. In fact, one can easily verify that the $g(t) = \frac{e^t}{1+e^t}$ is strictly monotone, and since $\mathcal{F}_1$ is 0-identifiable, so is $\mathcal{F}_2$. It remains to show that $a \mapsto D(\cdot; a)$ is robust.

(1) **[Lipschitz Optimal Price Mapping]** By calculation, we have $R'_a(x) = \frac{e(e^{ax}(1-ax)+e)}{(e^x+e)^2}$, and thus $R'_a(x) = 0$ iff $e^{ax}(1 - ax) + e = 0$, i.e., $-(1 - ax) = e^{1-ax}$. Letting $t = 1 - ax$, the above becomes

$$-t = e^t. \tag{4}$$

Note that RHS is increasing in $t$, while LHS is decreasing and surjective, so (4) admits a unique solution, say $t_0$. Recalling $x = \frac{1-t}{a}$, we obtain $p^*(a) = \frac{1-t_0}{a}$. Thus, the $p^*$ function is 2-Lipschitz since $a \geq \frac{1}{2}$.

(2) **[Robust Under Perturbation]** For any fixed $x \in [\frac{1}{2}, 1]$, we have $\Phi_x(a) = D(x; a) = \frac{e^{1-ax}}{1+e^{1-ax}}$. By calculation, for any $y \in \mathcal{R}(\Phi_x)$ we have

$$\Phi_x^{-1}(y) = \frac{1}{x} \left( 1 + \ln \frac{1-y}{y} \right),$$

and hence for any $y, y' \in \mathcal{R}(\Phi_x)$,

$$
\begin{aligned}
\left| \Phi_x^{-1}(y) - \Phi_x^{-1}(y') \right| &= \frac{1}{x} \cdot \left| \left( 1 + \ln \frac{1-y}{y} \right) - \left( 1 + \ln \frac{1-y'}{y'} \right) \right| \\
&= \frac{1}{x} \left| \ln y - \ln y' + \ln(1-y) - \ln(1-y') \right| \\
&\leq \frac{1}{x} \left( \max_{s \in \mathcal{R}(\Phi_x)} \frac{1}{s} \cdot |y - y'| + \max_{t \in \mathcal{R}(\Phi_x)} \frac{1}{1-t} \cdot |y - y'| \right) \\
&\leq \frac{|y - y'|}{x} \cdot \left( \max_{s \in \mathcal{R}(\Phi_x)} \frac{1}{s} + \max_{t \in \mathcal{R}(\Phi_x)} \frac{1}{1-t} \right), \tag{5}
\end{aligned}
$$

where the first inequality follows from the Lagrange Mean Value Theorem. Note that for any $a \in [\frac{1}{2}, 1]$ and $x \in [\frac{1}{2}, 1]$, we have

$$\frac{1}{2} \le \Phi_x(a) \le \frac{e^{\frac{3}{4}}}{1 + e^{\frac{3}{4}}} < \frac{7}{10},$$

so $\mathcal{R}(\Phi_x) \subset [\frac{1}{2}, \frac{7}{10}]$. Therefore,

$$(5) \le 2 \cdot |y - y'| \cdot \left(2 + \frac{10}{3}\right) = \frac{32}{3} \cdot |y - y'|,$$

and the proof completes. □

### A.5 Proof of Proposition 4

For any given $d$, consider $f(x) = \frac{\sin(d\pi x)}{d\pi}$ and $g(x) = \frac{\sin(2d\pi x)}{2d\pi}$ and $\mathbf{p} = (p_i) = (0, \frac{1}{d}, ..., \frac{d}{d})$. Observe that both $f$ and $g$ are 1-Lipschitz, and moreover, $f(p_i) = g(p_i) = 0$ for any $i$. Therefore, $\Phi_{\mathbf{p}}$ is not injective, and hence $\mathcal{F}$ is not $d$-identifiable. □

## B   Proofs of Upper Bounds

In this section, we prove the following tight regret bounds for the markdown version. To highlight the technical challenges, we first rephrase the known tight regret bound for non-markdown version.

**Theorem 8** ([3]). For any zero-dimensional demand family $\mathcal{F}$, there is an algorithm with regret $O(\log T)$. Moreover, there exists a zero-dimensional demand family $\mathcal{F}$ on which any algorithm has regret $\Omega(\log T)$.

Most of our upper bounds rely on the following standard concentration bound for subgaussian random variables (see e.g. [17]).

**Theorem 9** (Hoeffding's inequality). Suppose $X_1, .., X_n$ are independent subgaussian random variables, then for any $\delta > 0$,

$$\mathbb{P}\left[\sum_{i=1}^{n}(X_i - \mathbb{E}X_i) \ge \delta\right] \le \exp\left(-\frac{\delta^2}{2\sum_{i=1}^{n}\|X_i\|_{\psi_2}^2}\right).$$

We will also use a folklore result from Calculus.

**Theorem 10** (Taylor's Theorem with Lagrange Remainder). Let $f : \mathbb{R} \to \mathbb{R}$ be $(m+1)$ times differentiable on an open interval $(a, b)$. Then for any $x, x' \in (a, b)$, there exists some $\xi$ with $(x - \xi) \cdot (x' - \xi) \le 0$ such that

$$f(x') = f(x) + \frac{1}{1!}f'(x)(x' - x) + ... + \frac{1}{m!}f^{(m)}(x)(x' - x)^s + \frac{1}{(m+1)!}f^{(m+1)}(\xi)(x' - x)^{s+1}.$$

Theorem 10 implies a key property for any $s$-sensitive reward functions. Suppose $R$ is $s$-sensitive, then for any $\varepsilon > 0$, we have

$$R(p^* + \varepsilon) = R(p^*) + \frac{R^{(s)}(\xi)}{s!}\varepsilon^s$$

where $\xi \in [p^*, p^* + \varepsilon]$. Since $\Theta$ is compact, there exists some constant $C_s > 0$ such that $|R^{(s)}(x, \theta)| \le C_s$ for any $x \in [0, 1]$ and $\theta \in \Theta$. Thus,

$$|R(p^* + \varepsilon) - R(p^*)| \le \frac{C_s}{s!}|\varepsilon|^s.$$

Consequently, if a policy overshoots or undershoots the optimal price by $\varepsilon$, the regret *per round* is only $O(\varepsilon^s)$, which is asymptoticly lower than the per-round regret $O(\varepsilon^2)$ without the sensitivity assumption.

They considered a simple policy that estimates the true parameter using maximum likelihood estimator (MLE), and then selects the optimal price of the estimated demand function. To bound the expected regret in round $t$, they showed that the *mean squared error* (MSE) of the estimated price is at most $1/t$, and hence the expected total regret is $\sum_{t=1}^{T}\frac{1}{t} \sim \log T$.

## B.1 Zero-Dimensional Family

While Theorem 8 is established by bounding the Mean Square Error (MSE), due to the monotonicity constraint for markdown pricing, it no longer suffices to consider the *mean* error. Rather, we need an error bound which (i) holds with high probability, so that we can make *conservative* decision by selecting a price that is extremely unlikely to overshoot the optimal price, and (ii) is sufficiently low, so that the total regret is also low. The following lemma can be obtained as a direct consequence of Hoeffding's inequality (Theorem 9).

**Lemma 3.** Let $Z_1, .., Z_m$ be a i.i.d. samples from a distribution $D$ with subgaussian norm $C$. Let $\mathcal{B}$ be the event that $\left| \mathbb{E}[D] - \frac{1}{m} \sum_{j=1} Z_j \right| \leq 2C \cdot \sqrt{\frac{\log T}{m}}$, then $\mathbb{P}[\overline{\mathcal{B}}] \leq T^{-2}$.

*Proof.* By the Hoeffding inequality (Theorem 9), we have

$$
\mathbb{P}\left[ \left| \mathbb{E}[D] - \frac{1}{m} \sum_{j=1}^{m} Z_j \right| > 2C \sqrt{\frac{\log T}{m}} \right] \leq \exp\left( -\frac{(2C\sqrt{t_j \log T})^2}{2t_j \cdot C^2} \right) = T^{-2},
$$

and the proof follows. $\square$

Recall that $C_{sg}$ is the upper bound on the subgaussian norm of the demand distributions at any price, as formalized in Assumption 2, and $\bar{d}_j$ is the empirical mean demand in phase $j \in [K]$.

**Definition 12** (Good and Bad Events). For every $j \in [K]$, let $\mathcal{E}_j$ be the event that $\left| D(p_j; \theta^*) - \bar{d}_j \right| \leq 2C_{sg}\sqrt{\frac{\log T}{t_j}}$. We call $\mathcal{E} = \bigcap_{j=1}^{K} \mathcal{E}_j$ be the *good event* and its complement $\mathcal{E}^c$ the *bad event*.

As a standard step in regret analysis, we first show that the bad event occurs with low probability.

**Lemma 4** (Bad Event is Unlikely). $\mathbb{P}[\mathcal{E}^c] \geq 1 - T^{-1}$.

*Proof.* Fix any $j \in [m]$. Since $\{ D_j : j = t^{(j-1)} + 1, ..., t^{(j)} \}$ is an i.i.d. sample from a subgaussian distribution with mean $D(p_j; \theta^*)$ and subgaussian norm at most $C_{sg}$, by Lemma 3 we have $\mathbb{P}[\overline{\mathcal{E}_j}] \leq T^{-2}$. By the union bound, we have $\mathbb{P}[\mathcal{E}^c] \leq T^{-2} \cdot \log T \leq T^{-1}$. $\square$

Since the expected regret per round is at most $[0, 1]$, we can condition on $\mathcal{E}$ by losing only an $O(1)$ term in the regret.

**Lemma 5.** Conditional on $\mathcal{E}$, we have $\|\hat{\theta}_j - \theta^*\|_2 \leq 2C_2 \cdot C_{sg}\sqrt{\frac{\log T}{t_j}}$ for each $j \in [K]$.

*Proof.* Unless stated otherwise, we denote $\| \cdot \| = \| \cdot \|_\infty$. By definition robust parametrization,

$$
\|\hat{\theta}_j - \theta^*\| = \|\Phi_{p_j}^{-1}(\bar{d}_j) - \Phi_{p_j}^{-1}\left( \Phi_{p_j}(\theta^*) \right)\| \leq C_2 \cdot \|\bar{d}_j - \Phi_{p_j}(\theta^*)\| \leq 2C_2 \cdot C_{sg}\sqrt{\frac{\log T}{t_j}},
$$

where the last inequality follows since $\Phi_{p_j}(\theta^*) = D(p_j, \theta^*)$ and the assumption that $\mathcal{E}$ occurs. $\square$

We next show that the prices are guaranteed to be non-increasing conditional on the good event. To this aim, we introduce the following confidence intervals on the parameter space, centered at the estimated parameter $\hat{\theta}_j$.

**Definition 13** (Confidence Interval for $\theta^*$). Let $w_j = 4C_2 \cdot C_{sg}\sqrt{\frac{\log T}{t_j}}$. Define $L_j = \hat{\theta}_j - w_j$, $R_j = \hat{\theta}_j + w_j$, and $I_j = [L_j, R_j]$.

In our analysis of price monotonicity, we will show that $I_j$'s form a nested sequence. To this aim, we need to ensure that $\theta^*$ not only lies within $I_j$, but is also "far" from the boundary, in the sense that it lies between the first and third quantile of the interval. We formalize this notion as follows.

**Definition 14** (Far-From-the-Boundary). Given a finite interval $[a, b]$ and $x \in [a, b]$, we say $x$ is *Far-From-the-Boundary* (FFB) in $[a, b]$ if $\frac{3}{4}a + \frac{1}{4}b \leq x \leq \frac{1}{4}a + \frac{3}{4}b$.

Note that in Definition 13, we intentionally selected the radius to be *twice* as large as (rather than equal to) the bound given in Lemma 5, so that $\theta^*$ is FFB in $I_j$, conditional on the good event. This fact will be play a crucial role in the following analysis of monotonicity. We first observe that it suffices to show that $I_j$'s form a nested sequence.

**Lemma 6** (Nested Sequence of Intervals). For each $j$, we have $I_{j+1} \subseteq I_j$.

*Proof.* By symmetry, this is equivalent to that the right endpoints satisfy $R_{j+1} \leq R_j$. Recall that $\theta^*$ is FFB in $I_j$, so $R_j \geq \theta^* + \frac{1}{2}w_j$. Similarly, since $\theta^*$ is also FFB in $I_{j+1}$, we have $R_{j+1} \leq \frac{3}{2}w_{j+1} + \theta^*$. Combining, we have

$$R_{j+1} \leq \frac{3}{2}w_{j+1} + R_j - \frac{1}{2}w_j. \tag{6}$$

Finally, to bound the above by $R_j$ and hence complete the proof, it suffices to show $w_{j+1} \leq \frac{1}{3}w_j$. This is in fact straightforward from the choice of $w_j$. In fact, recall that $w_j = 4C_2 \cdot C_{sg}\sqrt{\frac{\log T}{t_j}}$ and $t_j = 9^j \log T$, so

$$\frac{w_{j+1}}{w_j} = \sqrt{\frac{t_j}{t_{j+1}}} = \sqrt{\frac{9^j \log T}{9^{j+1} \log T}} = \frac{1}{3},$$

and thus by (6), $R_{j+1} \leq R_j$, and the proof completes. $\qquad\square$

Recall that in Algorithm 1, to prevent the price from going up, in each phase $j$ we select $p_{j+1} = \min\{p_j, \tilde{p}_{j+1}\}$, where $\tilde{p}_{j+1} = \max\{p^*(\theta) : |\theta - \hat{\theta}_j| \leq w_j\}$ is called the *raw price*. We next show that such a truncation is almost "redundant", in the sense that under the good event the selected price $p_j$ is *always* the raw price.

**Proposition 5** (Just Select the Raw Price). Conditional on $\mathcal{E}$, we have $p_{j+1} = \tilde{p}_j$ for every $j$.

*Proof.* We show by induction that for any $j$, we have (i) $p_j = \max\{p^*(\theta) : |\theta - \hat{\theta}_j| \leq w_j\}$ and (ii) $\tilde{p}_{j+1} \leq p_j$. Note that (ii) implies that the policy selects $p_{j+1} = \tilde{p}_{j+1}$, and the proof follows.

For $j = 0$ this is trivially true. Now assume this holds for some $j \geq 1$, it then suffices to show that $\tilde{p}_{j+1} \leq p_j$. By Lemma 6, we have

$$\max\{p^*(\theta) : \theta \in I_{j+1}\} \leq \max\{p^*(\theta) : \theta \in I_j\}.$$

Further, note that by definition, we have $\tilde{p}_{j+1} = \max\{p^*(\theta) : \theta \in I_{j+1}\}$ and by induction hypothesis, we have $p_j = \max\{p^*(\theta) : \theta \in I_j\}$. Combining, we obtain $\tilde{p}_{j+1} \leq p_j$. $\qquad\square$

**Proof of Theorem 1.** As discussed earlier, since the bad event $\mathcal{E}^c$ occurs with $O(T^{-1})$ probability, we may preform the regret analysis assuming $\mathcal{E}$ occurs, by losing only an additive $O(1)$ term in the regret.

Since $\Phi$ is a robust parametrization, by definition we have

$$\begin{aligned}
\|\hat{\theta}_j - \theta^*\|_2 &= \|\Phi_{p_j}^{-1}(\bar{d}_j) - \Phi_{p_j}^{-1}\left(\Phi_{p_j}(\theta^*)\right)\| \\
&\leq C_2 \cdot \|\bar{d}_j - \Phi_{p_j}(\theta^*)\| \\
&= C_2 \cdot \|\bar{d}_j - D(p_j; \theta^*)\| \leq 2C_2 \cdot C_{sg}\sqrt{\frac{\log T}{t_j}}.
\end{aligned}$$

We next analyze $|p_{j+1} - p^*(\theta^*)|$. Note that by Proposition 5, $p_j = \tilde{p}_j = \max\{p^*(\theta) : \theta \in I_j\}$. By Assumption 5, the mapping $p^*$ is $C^*$-Lipschitz for some constant $C^* > 0$, so the price $p_{j+1}$ selected in the $(j+1)$-st phase satisfies

$$|p_{j+1} - p^*(\theta^*)| \leq C^*\|\hat{\theta}_j - \theta^*\|_2 \leq 2C_2 \cdot C^* \cdot C_{sg}\sqrt{\frac{\log T}{t_j}}.$$

Since the length of phase $j + 1$ is $t_{j+1}$, the regret incurred in this phase is at most $C_s \left(2C^* \cdot C_2 \cdot C_{sg} \sqrt{\frac{\log T}{t_j}}\right)^s \cdot t_{j+1}$ in expectation. Note that there are in total $K \leq \log T - \log \log T$ phases, so we can bound the cumulative regret as

$$\text{Reg(CM}, \mathcal{F}) \leq \sum_{j=1}^{K} C_s \left(2C^* \cdot C_2 \cdot C_{sg} \sqrt{\frac{\log T}{t_j}}\right)^s \cdot t_{j+1}$$

$$= C_s \left(2C^* \cdot C_2 \cdot C_{sg} \sqrt{\log T}\right)^s \cdot \sum_{j=0}^{K} \frac{t_{j+1}}{t_j^{s/2}} \tag{7}$$

We substitute $t_j$ with $\lceil 9^j \log T \rceil$ and simplify the above for $s = 2$ and $s > 2$ separately. When $s = 2$,

$$(7) = C_s \left(2C^* \cdot C_2 \cdot C_{sg} \sqrt{\log T}\right)^2 \cdot \sum_{j=0}^{K} \frac{t_{j+1}}{t_j}$$

$$\leq C_s \left(2C^* \cdot C_2 \cdot C_{sg}\right)^2 \cdot \log T \cdot 9(\log T - \log \log T)$$

$$= O(\log^2 T).$$

Now suppose $s > 2$. Then,

$$(7) = C_s \left(2C^* \cdot C_2 \cdot C_{sg} \sqrt{\log T}\right)^s \cdot \sum_{j=0}^{K} \frac{9^{j+1} \log T}{9^{j \cdot \frac{s}{2}} \log^{s/2} T}$$

$$\leq C_s \left(2C^* \cdot C_2 \cdot C_{sg}\right)^s \cdot \log T \cdot \sum_{j=0}^{K} 9^{(1-\frac{s}{2})j+1}$$

$$\leq 2C_s \cdot \left(2C_2 \cdot C^* \cdot C_{sg}\right)^s \cdot \log T \cdot \int_0^K 9^{(1-\frac{s}{2})x} dx$$

$$= 2C_s \cdot \left(2C^* \cdot C_2 \cdot C_{sg}\right)^s \cdot \log T \cdot \frac{2}{(s-2) \cdot \ln 9}$$

$$= O(\log T).$$

Theorem 1 follows by combining the analyses for $s = 2$ and $s > 2$. $\qquad\square$

## B.2 Finite-Dimensional Family

In this section we first analyze the regret of the ICM policy and prove Theorem 3, and then complement this upper bound with an almost matching lower bound. Recall that the ICM policy is specified by two types of parameters: the gap $h$ between neighboring sampling prices in each phase, and the number $T_j$ of rounds to stay at each sampling price in phase $j$. To prove Theorem 3, we first present the following upper bound on the regret of ICM for arbitrary choice of parameters $h$ and $T_j$'s, and then optimize the choice of parameters (up to polylogarithmic factors in $T$) by solving a linear program.

**Proposition 6.** Let $\mathcal{F}$ be a $d$-dimensional, $s$-sensitive ($s \geq 2$) family of demand functions. Suppose $h > 0$ and $0 < T_1 < ... < T_m$ where $T_m = o(T)$. Denote ICM = ICM$(T_1, ..., T_m, h)$. Then,

$$\text{Reg(ICM}, \mathcal{F}) \leq T_1 + C_s \left(2C^* C_{sg} h^{-d} \sqrt{C_5 d \log T}\right)^s \cdot \left(\sum_{j=1}^{m-1} T_{j-1}^{-s/2} \cdot T_j + T_m^{-s/2} \cdot T\right) + C_s \left(mdh\right)^s T.$$

We briefly explain the intuition behind the above result before proceeding with finding the optimal parameters. As the name suggests, the Iterative Cautious Myopic policy iteratively computes a confidence interval $[L_j, U_j]$ around the true optimal price, and conservatively moves to the right endpoint of this interval. As a *simplistic* view, in phase $j$ (assuming it ever takes place) the estimation error is $\sim h^{-d} T_{j-1}^{-1/2}$, and by definition of $s$-sensitivity, the regret incurred in phase $j$ is $\sim (h^{-d} T_{j-1}^{-1/2})^s T_j$.

To understand the final term, observe that when $h$ is sufficiently small compared to $U_j - L_j$, there is little risk of *overshooting* at the right endpoint $U_j$. However, when one selects larger $h$ (for faster

learning rate), it may happen that the last sample price $p_j - dh$ in this phase overshoots the optimal price, thereby incurring a regret term, as captured by the last term in the above bound.

Nonetheless, the actual proof involves carefully analyzing each of the three events (good, dangerous and overshooting) that can possibly occur at the end of each phase, as formally defined in Algorithm 2. Informally, each of these three events corresponds to the scenario where the price at the end of this phase lies (1) on the right, (2) inside, or (3) on the left of the confidence interval of the estimated optimal price.

**Lemma 7.** For each phase $j = 1, ..., m$, let $\mathcal{E}_j$ be the event that $p^*(\theta^*) \in [L_j, U_j]$. Then, $\mathbb{P}(\mathcal{E}_j) \geq 1 - dT^{-2}$.

*Proof.* By the Hoeffding inequality (Theorem 9) and the subgaussian assumption (Assumption 5), for each $k = 0, ..., d$, it holds with probability at least $1 - T^{-2}$ that

$$|D(p_i - kh; \theta^*) - \bar{d}| \leq 2C_{sg}\sqrt{\frac{\log T}{T_j}}.$$

For simplicity we write $\Phi = \Phi_{p_i, p_i - h, ..., p_i - dh}$ and $\bar{\mathbf{d}} = (\bar{d}_0, ..., \bar{d}_d)$. Since for any $v \in \mathbb{R}^d$ it holds $\|v\|_2 \leq \sqrt{d} \cdot \|v\|_\infty$, it holds with probability $1 - (d+1)T^{-2}$ that

$$\|\Phi(\theta^*) - \bar{\mathbf{d}}\|_2 \leq 2C_{sg}\sqrt{\frac{\log T}{T_j}} \cdot \sqrt{d}.$$

Thus by definition of dimension, for sufficiently large $T_j$ (hence sufficiently small $\|\Phi(\theta^*) - \bar{\mathbf{d}}\|_2$),

$$\begin{aligned}
\|\theta^* - \hat{\theta}\|_2 &= \|\Phi^{-1}(\bar{\mathbf{d}}) - \Phi^{-1}(\Phi(\theta^*))\|_2 \\
&\leq C_2 h^{-d} \cdot \|\Phi(\theta^*) - \bar{\mathbf{d}}\|_2 \\
&\leq C_2 h^{-d} \cdot C_{sg} 2\sqrt{\frac{\log T}{T_j}} \cdot \sqrt{d} = w_j,
\end{aligned}$$

and $\theta^* \in [L_j, U_j]$ follows immediately from the definition of $L_j$ and $U_j$. $\qquad\square$

**Proof of Proposition 6.** We first show that with high probability, our confidence interval forms a nested sequence of intervals containing the true parameter $\theta^*$. Recall that

$$L_j = \min\{p^*(\theta) : \|\theta - p^*(\hat{\theta})\|_2 \leq w_j\} \quad \text{and} \quad U_j = \max\{p^*(\theta) : \|\theta - p^*(\hat{\theta})\|_2 \leq w_j\}.$$

This lemma immediately implies a (high-probability) upper bound for the estimation error of the optimal price. Recall that $p_j = \max\{p^*(\theta) : \|\theta - p^*(\hat{\theta})\|_2 \leq w_j\}$. By Lemma 7 and Assumption 5, we deduce that conditional on $\mathcal{E}_j$, for any $p \in [L_j, U_j]$ it holds

$$|p - p^*(\theta^*)| \leq C^* \|\theta^* - \hat{\theta}\|_2 \leq C^* w_j.$$

We will repeatedly apply this bound in the following regret analysis.

**Proof of Proposition 6.** By Lemma 7,

$$\mathbb{P}(\bigcup_{i=1}^m \overline{\mathcal{E}_i}) \leq \sum_{i=1}^m \mathbb{P}(\overline{\mathcal{E}_i}) \leq dT^{-2} \cdot m \leq T^{-1}.$$

Thus, we may subsequently assume $\bigcap_{i=1}^m \mathcal{E}_i$ occurs by losing only an $O\left(\frac{1}{T}\right)$-factor in regret.

We split our proof into two cases depending on whether the overshooting event ever occurs in any phase.

**Case (1).** Suppose the overshooting event never occurs, i.e. in each $j = 1, ..., m$, we always have $p_j - dh \geq L_j \geq L_{j-1}$. Since $p_j \leq U_{j-1}$, we deduce that $p_j - kh \in [L_{j-1}, U_{j-1}]$ for all $k = 0, .., d$. On the other hand, since we have conditioned on $\bigcup_{i=1}^m \overline{\mathcal{E}_i}$, we have $p^*(\theta^*) \in [L_{j-1}, U_{j-1}]$, hence $|(p_j - kh) - p^*| \leq U_{j-1} - L_{j-1} \leq C^* w_j$ for $0 \leq k \leq d$. Thus the regret incurred in this phase is at

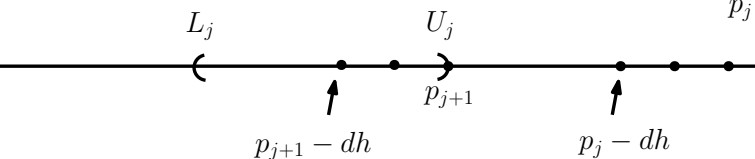

Figure 1: Illustration of case 2.

most $C_s \cdot (U_{j-1} - L_{j-1})^s T_j \leq C_s (C^* w_j)^s T_j$. Similarly, since the exploitation price $\tilde{p} := p_m - dh$ satisfies $|\tilde{p} - p^*(\theta^*)| \leq C^* w_m$, the expected regret per round in the exploitation phase is at most $(C^* w_m)^s$. Therefore, we may bound the cumulative regret as

$$\text{Reg}(\text{ICM}, \mathcal{F}) \leq T_1 + C_s (C^* w_1)^s T_2 + ... + C_s (C^* w_{m-1})^s T_m + C_s (C^* w_m)^s T$$

$$\leq T_1 + \sum_{j=2}^{m} C_s \left( C^* \cdot 2C_{sg} h^{-d} \sqrt{\frac{C_5 d \log T}{T_{j-1}}} \right)^s T_j + C_s \left( C^* \cdot 2C_{sg} h^{-d} \sqrt{\frac{C_5 d \log T}{T_m}} \right)^s T$$

$$= T_1 + C_s \left( 2C^* C_{sg} h^{-d} \sqrt{C_5 d \log T} \right)^s \cdot \left( \sum_{j=1}^{m-1} T_{j-1}^{-s/2} \cdot T_j + T_m^{-s/2} \cdot T \right).$$

**Case (2).** Now suppose the overshooting event first occurs in some phase $\ell$ where $1 \leq \ell \leq m$, formally

$$\ell = \min\{s : p_s - dh < L_j\}.$$

As in case (1), the expected regret in phase $j = 1, ..., \ell - 1$ can be bounded by $C_s \cdot (C^* w_{j-1})^s T_j$. Thus it remains to bound the expected regret in the $\ell$-th and the exploitation phase as $\tilde{O}((mdh)^s T)$. Suppose the last phase that good event occurred is phase $j$ (as illustrated in Fig 1). There are two sub-cases.

i) Suppose $j = \ell - 1$. Then, by definition of good event, we have $p_j - dh \geq U_j$. Thus, the ICM policy sets the next price to be $p_{j+1} = U_j$. Since $p^*(\theta^*) \in [L_j, U_j]$ and $p_\ell = p_{j+1} = U_j$, the exploitation price $p_\ell - dh$ satisfies

$$|p_\ell - dh - p^*(\theta^*)| \leq |p_\ell - dh - U_j| = |p_\ell - dh - p_\ell| = dh.$$

Thus, the future regret is at most $C_s(dh)^s T$.

ii) Now suppose $j \leq \ell - 2$. Then, the dangerous event must have occurred in phases $j + 1, j + 2, ... \ell - 1$, so

$$p_{j+s+1} = p_{j+s} - dh, \quad \forall s = 1, ..., \ell - j - 1.$$

In particular,

$$p_\ell = p_{j+1} - (\ell - j - 1) \cdot dh.$$

On the other side, by definition of the overshooting event, it holds

$$p_\ell - dh \leq L_{j+1} \leq p^*(\theta^*) \leq U_{j+1} \leq U_j = p_{j+1},$$

i.e. $p_\ell - dh \leq p^*(\theta^*) \leq p_{j+1}$. Thus,

$$|p_\ell - dh - p^*(\theta^*)| \leq (\ell - j - 1) \, dh.$$

Therefore, the regret in the exploitation phase is bounded by $C_s|p_\ell - dh - p^*|^s T \leq C_s(mdh)^s T$.

The proof completes by combining the analyses for the above cases.

We now determine the parameters to minimize the upper bound in Proposition 6.

**Proof of Theorem 3.** Write $T_i = T^{z_i}$, $h = T^{-y}$. Then for any $j \leq m - 1$,

$$(h^{-d} T_j^{-1/2})^s T_{j+1} = T^{sdy - \frac{s}{2}z_j + z_{j+1}}.$$

To find the optimal parameters, consider

$$\text{LP}(d): \quad \min_{x,y,z} \quad T^x$$

$$\text{subject to} \quad T^{z_1} \leq T^x, \quad\quad \text{Regret in phase 1}$$

$$T^{2sdy+z_2-\frac{s}{2}z_1} \leq T^x, \quad\quad \text{Regret in phase 2}$$

$$...$$

$$T^{sdy+1-\frac{s}{2}z_m} \leq T^x, \quad\quad \text{Regret in exploitation}$$

$$T^{1-sy} \leq T^x, \quad\quad \text{Regret for overshooting}$$

$$x, y, z \geq 0, z \leq 1$$

Taking logarithm with base $T$ on both sides, the above becomes

$$\min_{x,y,z} \quad x$$

$$\text{s.t.} \quad
\begin{bmatrix}
-1 & 0 & 1 & 0 & 0 & 0 & ... & 0 \\
-1 & sd & -\frac{s}{2} & 1 & 0 & 0 & ... & 0 \\
-1 & sd & 0 & -\frac{s}{2} & 1 & 0 & ... & 0 \\
-1 & sd & 0 & 0 & -\frac{s}{2} & 1 & ... & 0 \\
 & & & ...... & & & & \\
-1 & sd & 0 & 0 & 0 & ... & -\frac{s}{2} & 1 \\
-1 & sd & 0 & 0 & 0 & 0 & ... & -\frac{s}{2} \\
-1 & -s & 0 & 0 & 0 & 0 & ... & 0
\end{bmatrix}
\begin{bmatrix}
x \\ y \\ z_1 \\ z_2 \\ \vdots \\ \\ z_{m-1} \\ z_m
\end{bmatrix}
\leq
\begin{bmatrix}
0 \\ 0 \\ \vdots \\ 0 \\ -1 \\ -1
\end{bmatrix}$$

$$x, y, z \geq 0, \quad z \leq 1$$

Note that the above LP consists of $m + 2$ variables and $m + 2$ inequality constraints, so the minimum is attained when all inequalities become identities. In this case, we have

$$z_1 = x \tag{8}$$

$$z_2 - \frac{s}{2} z_1 = x - sdy$$

$$z_3 - \frac{s}{2} z_2 = x - sdy$$

$$...$$

$$z_m - \frac{s}{2} z_{m-1} = x - sdy$$

$$1 - \frac{s}{2} z_m = x - sdy$$

$$1 - sy = x \tag{9}$$

By telescoping sum, we have

$$1 - \left(\frac{s}{2}\right)^m z_1 = \left(1 + \frac{s}{2} + ... + \left(\frac{s}{2}\right)^{m-1}\right) \cdot (x - dsy).$$

Combining the above with (8) and (9), we have

$$1 + \left(1 + \frac{s}{2} + ... + \left(\frac{s}{2}\right)^{m-1}\right) d(1-x) = \left(1 + \frac{s}{2} + ... + \left(\frac{s}{2}\right)^m\right) x.$$

Rearranging, we obtain

$$x = \frac{1 + \left(1 + \frac{s}{2} + ... + (\frac{s}{2})^{m-1}\right) d}{\left(1 + \frac{s}{2} + ... + (\frac{s}{2})^{m-1}\right) \cdot (d+1) + \left(\frac{s}{2}\right)^m}.$$

In particular, for $s = 2$, the above becomes

$$x = \frac{md + 1}{m(d+1) + 1} = \frac{d}{d+1} + \frac{1}{m(d+1)^2}.$$

Choosing $m = \log T$, we have $T^x = \tilde{O}(T^{\frac{d}{d+1}})$.

## B.3 Infinite-Dimensional Family

In this section we first present a general regret upper bound for policy $\text{UE}_{\Delta,w}$, which immediately implies Theorem 5. To this aim, we need to introduce another constant $\eta$, as motivated by the the following result. For notational convenience, we abbreviate $\frac{\partial^k}{\partial x^k} R(x;\theta)$ as $R^{(k)}(x;\theta)$ for any $k \geq 0$.

**Lemma 8.** Let $\mathcal{F} = \{R(x,\theta) : \theta \in \Theta\}$ be a family of $s$-sensitive reward functions. Then, there exists a constant $\eta > 0$ such that for any $\theta \in \Theta$ and $x \in [p^*(\theta) - \eta, p^*(\theta)]$, it holds $R^{(s)}(x;\theta) < 0$ and

$$2R^{(s)}(p^*(\theta);\theta) \leq R^{(s)}(x;\theta) \leq \frac{1}{2}R^{(s)}(p^*(\theta);\theta).$$

*Proof.* First consider any fixed $\theta \in \Theta$. By definition of sensitivity, we have $R^{(1)}(p^*(\theta),\theta) = ... = R^{(s-1)}(p^*(\theta),\theta) = 0$ and $R^{(s)}(p^*(\theta),\theta) < 0$. Define

$$g(\theta) = \sup \left\{ \gamma \geq 0 \mid 2R^{(s)}(p^*(\theta);\theta) \leq R^{(s)}(x;\theta) \leq \frac{1}{2}R^{(s)}(p^*(\theta);\theta), \quad \forall x \in [p^*(\theta) - \gamma, p^*(\theta)] \right\}.$$

By continuity of $R^{(s)}$ in $x$, we have $g(\theta) > 0$ for any $\theta \in \Theta$. We complete the proof by showing that $\eta := \sup_{\theta \in \Theta} g(\theta) > 0$. Recall that $\Theta$ is compact, and $R^{(s)}$ is continuous in $\theta$, we know that $\eta$ can be attained, i.e., there exists some $\theta \in \Theta$ with $g(\theta) = \eta$. Moreover, note that for any $\theta$ we have $g(\theta) > 0$, therefore $\eta > 0$, and the proof completes. $\square$

We are now ready to state the main result in this section. Note that by choosing $\Delta = T^{-1/(3s+1)}$ and $w = T^{-2/(3s+1)}$, we immediately obtain Theorem 5.

**Proposition 7** (Upper Bound). Let $\mathcal{F}$ be any $s$-sensitive family for some $s \geq 2$. Suppose $\Delta \leq \frac{C_s}{8s!C^{(1)}}\eta^s$ and $m \geq 4$, then

$$\text{Reg}(\text{UE}_{\Delta,m}, \mathcal{F}) = O\left(\Delta^{-1}w^{-2}\log T + (w + \Delta^s)T\right)$$

where we recall that $w = 2C_{sg}\sqrt{\frac{\log T}{m}}$.

Our analysis proceeds by conditioning on the following the notion of clean event, which occurs with high probability as we will show soon.

**Definition 15** (Clean event). Let $\mathcal{E}_j$ be the event that $\left|R(x_j) - \bar{\mu}_j\right| \leq 2C_{sg}\sqrt{\frac{\log T}{m}}$, and $\mathcal{E} = \bigcap_{j=1}^{\lceil \Delta^{-1} \rceil} \mathcal{E}_j$.

Note that by our choice of $L_j, U_j$, we know that $\mathcal{E}$ is simply the event that $R(x_j) \in [L_j, U_j]$ for all $1 \leq j \leq \Delta^{-1}$. We next show that $\mathcal{E}$ occurs with high probability, and hence we may perform the analysis conditional on $\mathcal{E}$.

**Lemma 9.** $\mathbb{P}(\overline{\mathcal{E}}) \leq T^{-1}$.

*Proof.* Let $R$ be the true reward function. Recall that $Z_i^j$ for $i = t^{(j-1)} + 1, ..., t^{(j)}$ are i.i.d. samples from a subgaussian distribution with mean $R(x_j)$, and that Assumption 2 the sugaussian norm of this distribution is at most $C_{sg}$. Thus by Lemma 3, we have $\mathbb{P}[\overline{\mathcal{E}_j}] \leq T^{-2}$. By the union bound, we have $\mathbb{P}[\mathcal{E}] \geq 1 - T^{-2} \cdot \log T \geq 1 - T^{-1}$.

In the rest of this section we will fix a true reward function $R(x;\theta)$ and write $x^* = p^*(\theta)$ and $R(x) = R(x;\theta)$.

**Definition 16.** Define $x_\ell$ be the closest sample price to $x^*$, i.e. $\ell := \arg\min_{0 \leq j \leq \Delta^{-1}} \{|x_j - x^*|\}$.

We first show that conditional on $\mathcal{E}$, the policy will stop reducing the price and enter the exploitation phase before reaching $x^* - \eta$.

**Lemma 10.** Suppose $\mathcal{E}$ occurs. For any $m \geq \left( \frac{4s! \cdot 8 \cdot C_{sg}}{C_s} \right)^2 \cdot \eta^{-2s} \log T$ and $\Delta \leq \frac{C_s}{8s! C^{(1)}} \eta^s$, we have $x_h \geq x^* - \eta$.

*Proof.* Recall that $x$ is said to be a sample price if $x = 1 - j\Delta$ for some integer $j$. Consider the smallest sample price $\tilde{x}$ above $x^* - \eta$, then $|x^* - \eta - \tilde{x}| \leq \Delta$. By Assumption 4, the first derivatives are bounded by $C^{(1)}$ and hence $R$ is $C^{(1)}$-Lipschitz, so

$$|R(\tilde{x}) - R(x^* - \eta)| \leq C^{(1)}|x^* - \eta - \tilde{x}| \leq C^{(1)}\Delta \quad \text{and} \quad |R(x_\ell) - R(x^*)| \leq C^{(1)}\Delta.$$

Moreover, by Theorem 10, and since $R^{(1)}(x^*) = ... = R^{(s-1)}(x^*) = 0$,

$$|R(x^* - \eta) - R(x^*)| = \left| \frac{R^{(s)}(\xi)}{s!} \eta^s \right| \tag{10}$$

for some $\xi \in (x^* - \eta, x^*)$. By Lemma 8, $|R^{(s)}(\xi)| \geq \frac{1}{2} \cdot |R^{(s)}(x^*)|$, so

$$|R(x^* - \eta) - R(x^*)| \geq \frac{|R^{(s)}(x^*)|}{2s!} \eta^s. \tag{11}$$

By combining the inequalities (10) and (11), we have

$$R(\tilde{x}) \leq R(x_\ell) - \left( \frac{|R^{(s)}(x^*)|}{2s!} \eta^s - 2C^{(1)}\Delta \right).$$

Recall that $|R^{(s)}(x^*)| \geq C_s$, so for any $\Delta \leq \frac{C_s}{8s! C^{(1)}} \eta^s$, we have

$$\frac{|R^{(s)}(x^*)|}{2s!} \eta^s - 2C^{(1)}\Delta \geq \frac{|R^{(s)}(x^*)|}{4s!} \eta^s. \tag{12}$$

Hence, suppose $m \geq \left( \frac{4s! \cdot 8 \cdot C_{sg}}{C_s} \right)^2 \cdot \eta^{-2s} \log T$, then $4w \leq \frac{C_s}{4s!} \eta^s \leq \frac{|R^{(s)}(x^*)|}{4s!} \eta^s$, and by (12)

$$R(\tilde{x}) < R(x_\ell) - 4w. \tag{13}$$

Since $\mathcal{E}$ occurs, we have $|U(\tilde{x}) - R(\tilde{x})| \leq w$ and $|L(x_\ell) - R(x_\ell)| \leq w$. Combining with (13), we obtain $U(\tilde{x}) < L(x_\ell)$, and thus the halting criterion is satisfied at $\tilde{x}$, so $x_h \geq x^* - \eta$.

**Lemma 11.** Suppose $x_{k+\ell} \geq x^* - \eta$. Then,

$$|R(x_{k+\ell}) - R(x^*)| \geq \frac{R^{(s)}(x^*)}{2s!} ((k-1)\Delta)^s.$$

*Proof.* By Theorem 10, $|R(x_{k+\ell}) - R(x^*)| = \left| \frac{1}{s!} R^{(s)}(\xi) \cdot (x_{k+\ell} - x^*)^s \right| \geq \frac{1}{2s!}|R^{(s)}(x^*)| \cdot |x_{k+\ell} - x^*|^s$. By definition of $x_\ell$, we have $|x_\ell - x^*| \leq \Delta$, so $|x^* - x_{k+\ell}| \geq |x_\ell - x_{k+\ell}| - |x^* - x_\ell| \geq (k-1)\Delta$. Thus, $|R(x_{k+\ell}) - R(x^*)| \geq \frac{1}{2s!}|R^{(s)}(x^*)| \cdot ((k-1)\Delta)^s$.

**Lemma 12.** Suppose $\mathcal{E}$ occurs and $i := \arg\max_{0 \leq j \leq \Delta^{-1}}\{L_j\}$. Then,

$$|R(x_i) - R(x_\ell)| \leq \max\{2w, \frac{2R^{(s)}(x^*)}{s!}\Delta^s\}.$$

*Proof.* Since $|x_\ell - x^*| \leq \Delta \leq \eta$, by Theorem 10, it holds that $R(x_\ell) \geq R(x^*) - \frac{2R^{(s)}}{s!}|x^* - x_\ell|^s \geq R(x_i) - \frac{2R^{(s)}}{s!}|x^* - x_\ell|^s$, and thus $R(x_i) - R(x_\ell) \leq \frac{2R^{(s)}}{s!}|x^* - x_\ell|^s \leq \frac{2R^{(s)}}{s!}\Delta^s$. On the other hand, by definition of $i$, we have $L_\ell \leq L_i$. Since $\mathcal{E}$ occurs, we also have $|L_i - R(x_i)| \leq w$ and $|L_\ell - R(x_\ell)| \leq w$, and therefore $R(x_\ell) - R(x_i) \leq 2w$, and the proof follows. $\square$

**Lemma 13.** Suppose $\mathcal{E}$ occurs and $\Delta \leq \eta$. Then for any $k \geq 3$ with $x_{k+\ell} \geq x^* - \eta$,

$$|R(x_{\ell+k}) - R(x_i)| \geq \frac{C_s}{2^s s!} (k\Delta)^s - 2w.$$

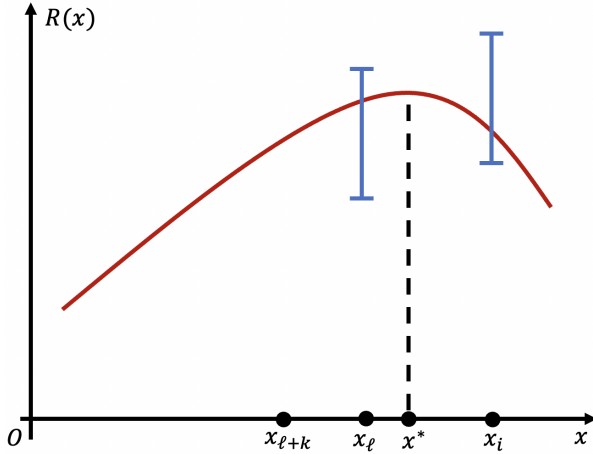

Figure 2: Illustration of Lemma 14

*Proof.* By Lemma 11, 12 and the triangle inequality,

$$|R(x_{\ell+k}) - R(x_i)| \geq |R(x_{\ell+k}) - R(x^*)| - |R(x^*) - R(x_\ell)| - |R(x_i) - R(x_\ell)|$$

$$\geq \frac{C}{s!} \cdot ((k-1)\,\Delta)^s - \frac{C_s}{s!}\Delta^s - 4w + \frac{C_s}{s!}\Delta^s$$

$$\geq \frac{C_s}{s!}\Delta^s \left((k-1)^s - 1\right) - 4w$$

$$\geq \frac{C_s}{2^s s!}(k\Delta)^s - 4w,$$

and the proof follows. $\square$

The crux for proving our lower bound lies in analyzing the regret in the exploitation phase. To this aim, we use the above lemma to bound the per-round regret in the exploitation phase, formally stated below, and illustrated in Figure 2. Recall that $x_h$ is the halting price, which is selected in every round in the exploitation phase.

**Lemma 14.** Suppose $\mathcal{E}$ occurs, then $R(x_h) - R(x^*) \leq \frac{3^s s! \cdot 4w}{C_s} + \max\{3^s, C_s\} \cdot \Delta^s$.

*Proof.* Consider two cases regarding $h$ and $\ell$.

**Case 1.** Suppose $h \geq \ell - 2$, i.e. $x_h \geq x_\ell - 2\Delta$. Since $|x_\ell - x^*| \leq \Delta$, we have $|x_h - x^*| \leq |x_\ell - x_h| + |x^* - x_\ell| \leq 2\Delta + \Delta = 3\Delta$. Thus by definition of sensitivity, when $|x_h - x^*| \leq \eta$ it holds $|R(x^*) - R(x_h)| \leq C_s \cdot |x_h - x^*|^s \leq C_s \cdot 3^s \Delta^s$.

**Case 2.** Now suppose $h \leq \ell - 3$, i.e. $x_h \leq x_\ell - 3\Delta$. Let $k = h - \ell - 1$, so that $x_{\ell+k}$ is the last sample price that the UE policy selected before halting at $x_h$. Then by definition of $x_h$, the halting criterion is *not* satisfied at the $x_{\ell+k}$, i.e. $[L(x_i), U(x_i)] \cap [L(x_{\ell+k}), U(x_{\ell+k})] \neq \emptyset$, so $|R(x_i) - R(x_{\ell+k})| \leq 4w$. Combining with Lemma 13, we have $2w \geq |R(x_i) - R(x_{\ell+k})| \geq \frac{C_s}{2^s s!}(k\Delta)^s - 2w$, i.e.,

$$(k\Delta)^s \leq \frac{2^s s! \cdot 4w}{C_s}. \tag{14}$$

Note that $|x_h - x^*| \leq (k+1)\Delta$, and recall that conditional on $\mathcal{E}$, we have $x_h \geq x^* - \eta$, it follows that

$$|R(x_h) - R(x^*)| \leq C_s \cdot ((k+1)\Delta)^s \qquad\qquad \text{by Theorem 10}$$

$$\leq C_s \cdot \left(\frac{3}{2}k\Delta\right)^s \qquad\qquad \text{since } k = h - \ell - 1 \geq 2$$

$$\leq \frac{4w \cdot 3^s s!}{C_s} = \frac{4 \cdot 3^s s! \cdot w}{C_s}, \qquad\qquad \text{by (14)}$$

and the proof is complete. $\square$

**Proof of Proposition 7.** Fix any $R \in \mathcal{F}$. Suppose $\mathcal{E}$ does not occur, then the regret is at most $T$. Suppose $\mathcal{E}$ occurs, then by Lemma 14, the regret incurred in the exploitation phase is bounded by $\left( \frac{6 \cdot C_s \cdot 3^s s!}{C_6} \cdot w + \max\{3^s, C_s\} \cdot \Delta^s \right) T$.

On the other side, recall that each sample price is selected for at most $m$ times, so the cumulative regret incurred in the exploration phase is bounded by $mT$. Moreover, there are at most $\lceil \Delta^{-1} \rceil$ sample prices. Recalling that $w = 2C_{sg}\sqrt{\frac{\log T}{m}}$, i.e. $m = 4C_{sg}^2 w^{-2} \log T$, we can bound the total regret as

$$
\begin{aligned}
& \mathrm{Reg}(\mathrm{UE}_{\Delta, w}, R) \\
& \leq \mathbb{P}[\overline{\mathcal{E}}] \cdot T + \mathbb{P}[\mathcal{E}] \cdot \left( 4C_{sg}^2 w^{-2} \Delta^{-1} \log T + \left( \frac{6 \cdot C_s \cdot 3^s s!}{C_6} \cdot w + \max\{3^s, C_s\} \cdot \Delta^s \right) \cdot T \right) \\
& \leq T^{-1} \cdot T + \left( 4C_{sg}^2 w^{-2} \Delta^{-1} \log T + \left( \frac{6 \cdot C_s \cdot 3^s s!}{C_6} \cdot w + \max\{3^s, C_s\} \cdot \Delta^s \right) \right) \cdot T \\
& = O\left( \Delta^{-1} w^{-2} \log T + (\Delta^s + w)T \right),
\end{aligned}
$$

and Proposition 7 follows. $\qquad\square$

## C   Proofs of Lower Bounds

### C.1   Preliminaries

We now turn to proving our lower bound, which establishes minimax optimality. Our proof considers *Bernoulli* reward distribution at each price and employs the following alternate view of a *policy* as binary decision trees, which we will make precise in this section.

**Definition 17** (Prefix). Let $\{0,1\}^* = \bigcup_{n=1}^{\infty} \{0,1\}^n \cup \{\mathrm{null}\}$ be the set of all finite-length binary vectors, where *null* denotes the empty binary vector. For any $v \in \{0,1\}^*$ and $k \in \mathbb{Z}$, the *k-prefix* of $v$ is defined as $v^k = (v_1, ..., v_k)$.

We will consider probability spaces on sets containing the prefixes of every element.

**Definition 18** (Downward Closed Sets). For any $v, w \in \{0,1\}^*$, we define $w \prec v$ if there exists $k \in \mathbb{Z}$ such that $v^k = w$. A set $\Omega \in \{0,1\}^*$ is *downward closed*, if for any $v \in \Omega$ and $w \prec v$, we have $w \in \Omega$.

A decision tree is specified by a downward closed set equipped with a real-valued function.

**Definition 19** (Decision Tree). A *binary decision tree* is a tuple $(\Omega, x)$ where $\Omega \subseteq \{0,1\}^*$ is downward closed and $x : \Omega \to \mathbb{R}$ is a mapping. Moreover, each $v \in \Omega$ is called a *node*.

Intuitively, for each node $v = (v_1, ..., v_k)$, the value $x(v)$ is just the *price* that the policy selects upon observing demands $v_1, ..., v_k$ at prices $x(v^1), ..., x(v^k)$. Recalling that we have normalized the price space to be $[0, 1]$, so we will subsequently consider only decision trees $(\Omega, x)$ with $0 \leq x(v) \leq 1$ for all $v \in \Omega$. For notational convenience, we suppress the notation $x(v)$ simply as $x_v$.

We next introduce an equivalent definition of markdown policy, using the language of decision tree.

**Definition 20** (Markdown Policy, Equivalent Definition). A *markdown policy* is a decision tree $(\Omega, x)$ such that it holds that $x(v^1) \geq x(v^2) \geq ... \geq x(v^k)$ for any $v = (v_1, ..., v_k) \in \Omega$.

We next introduce some standard terminologies for decision trees, in case the reader is not familiar with graph theory.

**Definition 21** (Decision Tree Basics). Let $\mathbb{A} = (\Omega, x)$ be a decision tree and $v, w \in \Omega$.

  i). We say $v$ is a *leaf* if there does not exist $w \in \Omega$ with $v \prec w$.

  ii). The *depth* $d(v)$ of $v$ is defined to be the length of binary vector $v$. Denote $L(\Omega) \subseteq \Omega$ the subset of all leaves. Each node in $\Omega \backslash L(\Omega)$ is called an *internal* node.

iii). We say $w$ is an *ancestor* of $v$ if $w \prec v$. If in addition, $d(v) = d(w) + 1$, then we say $w$ is the *parent* of $v$ and denote $w = par(v)$, and say $v$ is a *child* of $w$.

iv). A decision tree is *binary* if every internal has exactly two children.

Given a binary decision tree, every reward function induces a natural probability measure over the leaves. In fact, consider a random walk from the root to a random leaf, where at each internal node $v$, the walk moves to each of the two children with probability $R(x_v)$ and $1 - R(x_v)$ respectively, corresponding to whether there is a unit demand in this round. We formally define this probability measure below.

**Definition 22** (Probability Measure on Leaves). Let $(\Omega, x)$ be a decision tree and $R : [0, 1] \to [0, 1]$. Write $L = L(\Omega)$. For each $\ell = (\ell_1, ..., \ell_d) \in L$, define

$$p_R(\ell) = \prod_{j=1}^{d} R\left(x(\ell^j)\right)^{\ell_j} \cdot \left(1 - R\left(x(\ell^j)\right)\right)^{1-\ell_j}$$

The probability measure $\mathbb{P}_R$ on $(L, 2^L)$ is then given by $\mathbb{P}_R(S) = \sum_{\ell \in S} p_R(\ell)$ for each $S \subseteq L$. We also define $\mathbb{E}_R$ to be the expectation under the probability measure $\mathbb{P}_R$.

All of our lower bounds rely upon the following lower bound of the sample complexity for distinguishing between two demand models using an adaptive classifier, due to Wald and Wolfowitz [18]. Intuitively, an adaptive classifier is an algorithm that adaptively collects samples at possibly different prices from a fixed but unknown demand model, until certain stopping criterion is satisfied, wherein the algorithm returns one demand model from a set of two candidates $R$ and $B$ (referred to as "red" and "blue"). We first formally define the confidence of a classifier.

**Definition 23** (Adaptive Classifier). Consider functions $R, B : [0, 1] \to [0, 1]$. Let $(\Omega, x)$ be a decision tree and $f : L(\Omega) \to \{R, B\}$ be a mapping. The tuple $(\Omega, x, f)$ is then called an *adaptive classifier* for $R$ and $B$. Moreover, given $\alpha, \beta \in [0, 1]$, an adaptive classifier $(\Omega, x, f)$ is called $(\alpha, \beta)$-*confident* if $\mathbb{P}_R\left(f^{-1}(R)\right) \geq \alpha$ and $\mathbb{P}_B\left(f^{-1}(R)\right) \leq \beta$.

We may also refer to $\mathbb{P}_R\left(f^{-1}(B)\right)$ and $\mathbb{P}_B\left(f^{-1}(R)\right)$ as the type I and type II error. Then, an adaptive classifier is $(\alpha, \beta)$-confident if and only if the type I and type II errors are at most $1 - \alpha$ and $\beta$ respectively.

The following key result states that any adaptive classifier achieving a given confidence level must query at least a certain number of samples in expectation.

**Theorem 11** ([18]). Consider $R, B : [0, 1] \to [0, 1]$ and an $(\alpha, \beta)$-confident adaptive classifier $(\Omega, x, f)$. Denote $\Delta(R, B) = \max_{v \in \Omega} \text{KL}\left(R(x_v), B(x_v)\right)$ and let $D(\ell)$ be the depth of leaf $\ell \in L(\Omega)$. Then,

$$\mathbb{E}_R[D] \geq \frac{\alpha \log \frac{\alpha}{\beta} + (1 - \alpha) \log \frac{1-\alpha}{1-\beta}}{\Delta(R, B)} \quad \text{and} \quad \mathbb{E}_B[D] \geq \frac{\beta \log \frac{\beta}{\alpha} + (1 - \beta) \log \frac{1-\beta}{1-\alpha}}{\Delta(B, R)}. \tag{15}$$

For example, when $\alpha = \frac{3}{4}, \beta = \frac{1}{4}$, the above becomes $\frac{1}{\Delta(R,B)}$ and $\frac{1}{\Delta(B,R)}$ respectively, and the lower bounds scale inverse proportionally to the KL-divergence.

## C.2 Zero-Dimensional Family

We will use the *contrapositive* of Theorem 11.

**Corollary 2.** Let $R, B : [0, 1] \to [0, 1]$, $\alpha, \beta \in [0, 1]$ be constants. Suppose $(\Omega, x, f)$ be an $(\alpha', \beta')$-confident adaptive classifier for $R$ and $B$ with $\beta' \leq \beta$, such that at least one of the following holds:
(i) $\mathbb{E}_R[D] < \frac{\alpha \log \frac{\alpha}{\beta} + (1-\alpha) \log \frac{1-\alpha}{1-\beta}}{\Delta(R,B)}$,
(ii) $\mathbb{E}_B[D] < \frac{\beta \log \frac{\beta}{\alpha} + (1-\beta) \log \frac{1-\beta}{1-\alpha}}{\Delta(B,R)}$.
Then, $\alpha' \leq \alpha$.

In particular, we will choose $\alpha = \frac{3}{4}$ and $\beta = \tilde{O}(T^{-1/2})$ on a suitably constructed adaptive classifier. In this case, the above corollary says that if an adaptive classifier has type II error $\tilde{O}(T^{-1/2})$ and, in addition, at least one of $\mathbb{E}_R[D]$ and $\mathbb{E}_B[D]$ is $O(\sqrt{T})$, then the type I error is at *least* $\frac{1}{4}$ which, as we will soon see, leads to high regret.

Consider the family of linear demand function $\mathcal{F} = \{D(p; \theta) : \theta \in [1, 2]\}$ where $D(p; \theta) = 1 - \theta p$ for any $p \in [0, 1]$. We leave it to the reader to verify that $d(\mathcal{F}) = 1$. We will apply Corollary 2 on the following pairs of demand functions. Let $R(p) = 1 - p$, whose optimal price is $p_R^* = \frac{1}{2}$. We construct a *blue* demand function for each fixed $t \in [T]$. Let $\Delta_t = \sqrt{\frac{\log T}{t}}$ and define $B_t(p) = 1 - (1 - 2\Delta_t)p$. Then, the optimal price of $B_t$ is

$$p_{B_t}^* = \frac{1}{2(1 - 2\Delta_t)} = \frac{1}{2} + (1 + o(1)) \cdot \Delta_t,$$

where the second equality follows since $\frac{1}{1-\varepsilon} = 1 + \varepsilon + o(\varepsilon)$ as $\varepsilon \to 0$. It is straightforward then to verify that the following.

**Observation 1.** If $\log T \le t < \sqrt{T}$, then $p_R^* + \Delta_t \le p_{B_t}^* \le p_R^* + 2\Delta_t$.

The following lemma says that the KL-divergence from $R(p)$ to $B(p)$ is small at every price $p$, and hence it is hard to distinguish between $R$ and $B_t$.

**Lemma 15.** Let $\Delta(R, B) = \max_{p \in [0,1]} \mathrm{KL}(R(p), B_t(p))$. Then, $\Delta(R, B) \le 16\Delta_t^2$.

*Proof.* Note that if $X, Y$ are Bernoulli random variables with means $q$ and $q + \delta$ respectively, then $\mathrm{KL}(X, Y) \le \left( \frac{1}{q} + \frac{1}{1-q} \right) \cdot \delta^2$. Note that by our construction, for any $p \in [0, 1]$ we have $B_t(p), R(p) \ge \frac{1}{2}$ and

$$B_t(p) - R(p) = (1 - (1 - 2\Delta_t)p) = 2\Delta_t \cdot p - (1 - p) \le 2\Delta_t,$$

so $\mathrm{KL}(R(p), B_t(p)) \le 4 \cdot (2\Delta_t)^2 = 16\Delta_t^2$, and the proof is complete. $\square$

Subsequently, we will fix some $t \in [\log T, \sqrt{T})$ and denote $B = B_t$ for simplicity. To apply Corollary 2, consider the following adaptive classifier $(\Omega_t, x_t, f_t)$. Let $\pi = (\Omega, x)$ be any markdown policy where we recall that $\Omega$ are all the nodes of $\pi$ when viewed as a decision tree, and $x : \Omega \to [0, 1]$ represents the price selected at each node. Define $\Omega_t$ to be the nodes in $\Omega$ with depth at most $t$, and define $x_t = x|_{\Omega_t}$, i.e., the mapping $x : \Omega \to [0, 1]$ restricted to $\Omega_t$. Moreover, define $f_t : L(\Omega_t) \to \{R, B\}$ where

$$f_t(\ell) = \begin{cases} B, & \text{if } x(\ell) > p_R^* + \frac{\Delta_t}{2}, \\ R, & \text{else.} \end{cases}$$

For simplicity we will also denote $f = f_t$. To apply Corollary 2, with some foresight we choose $\alpha' = \mathbb{P}_R[f^{-1}(B)], \beta' = \mathbb{P}_B[f^{-1}(R)], \alpha = \frac{3}{4}$ and $\beta = T^{-1/2} \log T$.

**Lemma 16.** $\beta' \le \beta$.

*Proof.* Suppose the true demand function is $B$ and consider $\ell \in f^{-1}(R)$. By definition of $f$, we know that $\pi$ selects a price lower than $p_B^* - \Delta_t/2$. Then, due to the markdown constraint, and by Observation 1, an $\Omega(\Delta_t^2)$ regret is incurred in each future round and hence the cumulative regret is $\Omega(\beta' \Delta_t^2 T)$. Thus, if $\pi$ has $O(\log^2 T)$ regret, then $\beta' \Delta_t^2 T \le \log^2 T$, i.e., $\beta' \le \frac{t \log T}{T} = T^{-1/2} \log T = \beta$. $\square$

**Proof of Theorem 2.** Suppose $\mathrm{Reg}(\pi; \mathcal{F}) \le \log^2 T$. To apply Corollary 2, we choose $\alpha' = \mathbb{P}_R[f^{-1}(B)], \beta' = \mathbb{P}_B[f^{-1}(R)], \alpha = \frac{3}{4}$ and $\beta = T^{-1/2} \log T$. Then,

$$\frac{\alpha \log \frac{\alpha}{\beta} + (1 - \alpha) \log \frac{1-\alpha}{1-\beta}}{\Delta(R, B)} = \frac{16 \log T}{\Delta(R, B)}.$$

Recall that in the adaptive classifier $(\Omega_t, x_t, f_t)$, the depth $D$ of every leaf is *exactly* $t$, so $\mathbb{E}_R[D] = t$. Since $t < \sqrt{T}$, by Lemma 15 we obtain

$$\mathbb{E}_R[D] = t < \frac{\log T}{\Delta_t^2} \leq \frac{16 \log T}{\Delta(R, B)}.$$

Moreover, by Lemma 16, we deduce that $\beta' \leq \beta$. Therefore by Corollary 2, we have $\alpha' \leq \alpha$, i.e., $1 - \alpha' \geq 1 - \alpha = \frac{1}{4}$. In other words,

$$\mathbb{P}_R[f^{-1}(B)] = \mathbb{P}_R[x_t \geq p_R^* + \frac{\Delta_t}{2}] \geq \frac{1}{4}.$$

We conclude the proof by analyzing the regret in round $t$. Observe that by definition of $R$, for any $p \geq p_R^* + \frac{\Delta_t^2}{2}$ we have $R(p) \leq R(p_R^*) - \Delta_t^2$. Hence, the expected regret in round $t$ is at least $\frac{1}{4} \cdot \Delta_t^2 = \frac{\log T}{4t}$. Finally, note that the above argument holds for all $t \in [\log T, T^{1/2})$, we have

$$\text{Reg}(\pi, R) \geq \sum_{t=\log T}^{\sqrt{T}-1} \frac{\log T}{4t}$$

$$= \frac{\log T}{4} \cdot \left( \sum_{t=1}^{\sqrt{T}-1} \frac{1}{t} - \sum_{t=1}^{\log T} \frac{1}{t} \right)$$

$$= \Omega(\log^2 T) - O(\log T \cdot \log \log T) = \Omega(\log^2 T),$$

and the proof is complete. $\qquad\square$

### C.3 Finite-Dimensional Family

We first describe intuition behind the proof (see Figure 3). For each $d$ we construct a pair of demand functions $D_b, D_r$ on price space $[0, 1]$ with $D_b(1) = D_r(1)$. Moreover, price 1 is the unique optimal price of $D_b$ and suboptimal for $D_r$. Since the gap between these two demand functions is very small near price 1, to distinguish between them the policy has to reduce explore prices sufficiently lower than 1. However, if it reduces the price by too much, a high regret is incurred under $D_b$ since its optimal price is at 1.

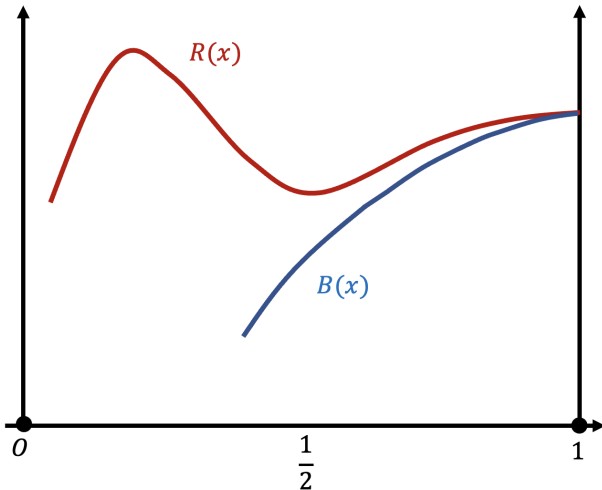

Figure 3: Illustration of Lemma 17.

We now sketch the proof at a high level before presenting the details. Consider a policy $\pi$. Choosing a suitable small number $h > 0$, we convert $\pi$ into an adaptive classifier for $D_r$ and $D_b$, based on whether the price in round $\frac{T}{4}$ is higher than $1 - h$. We first argue that if $\pi$ admits low regret, then this adaptive classifier must be $(\Omega(1), O(1))$-confident, otherwise an $\Omega(h^2 T)$ regret is incurred. Then

by Theorem 11, to distinguish between $D_r$ and $D_b$, $\pi$ has to select prices in $[1 - h, h]$ for $\Omega(h^{-2d})$ rounds in expectation, incurring $\Omega(h^{-2d} \cdot T)$ regret under $R_r$.

Now we formalize the above ideas. We first explicitly construct a pair of degree-$d$ polynomial demand functions that are hard to distinguish between.

**Lemma 17.** For any $d \geq 1$, there exists a pair $D_r, D_b$ of degree-$d$ polynomial demand functions satisfying the following properties.
(1) **Monotonicity:** Both are non-increasing on $[\frac{1}{2}, 1]$,
(2) **First-Order Optimality:** Denote $R_i(x) = x \cdot D_i(x)$ for $i \in \{r, b\}$. Then, the maximum of $R_b(x)$ is attained at $x = 1$. Moreover, $R'_b(1) = 0$,
(3) **Interior Optimal Price:** The function $R_r$ is maximized at some price $x \in [0, \frac{1}{2}]$,
(4) **Hardness of Testing:** Let $Gap(h) = \max_{x \in [1-h,1]}\{|D_r(x) - D_b(x)|\}$, then $Gap(h) \leq O(h^d)$ as $h \to 0^+$. In particular, this implies that $R_b(1) = R_r(1)$.

*Proof.* The proof involves explicit construction of the desired families of demand functions. In the next two subsections, we consider the case $d = 1$ and $d \geq 2$ separately.

**Step 1.** Suppose $d = 1$. Let $p_{min} = \frac{1}{2}, p_{max} = 1$. Consider the demand functions
$$D_b(1 - h) = 1 + h, \quad D_r(1 - h) = 1 + 5h.$$

Equivalently, substituting $x = 1 - h$, we have
$$D_b(x) = 2 - x, \quad D_r(x) = 6 - 5x.$$

Let us verify each of the four conditions in Lemma 17:
(1) both curves are clearly strictly decreasing.
(2) $R_0(x) = x(2 - x)$, so $R'_0(x) = 2 - 2x$. So its unique local maximum is attained at $x = 1$. Moreover, $R''_0(1) = -2 < 0$.
(3) $R'_1(x) = 6 - 10x$, so $R_1$ attains maximum at $x = \frac{3}{5}$.
(4) $|D_b(1 - h) - D_r(1 - h)| = 4 - 4h = O(h)$.

**Step 2.** Suppose $d \geq 2$. In this case, consider the following two demand functions:
$$D_b(M - h) = 1 + h + bh^d, \quad D_r(M - h) = 1 + h + rh^d,$$

defined on the interval $[0, M]$ where $M$ will be chosen to be some large number soon. The proof then follows by replacing $h$ with $Mh$, hence re-scaling the domain to $[0, 1]$.

We first verify some trivial properties. Note that $D_b(1 - h) - D_r(1 - h) = (r - b)h^d$, so the gap between these two demand functions around price $M$ is on the order of $O(h^d)$, and hence the last condition is satisfied.

We next verify that when $b = M^{-d}$, the function $R_b(x)$ attains maximum at $x = M$, formally, for any $d \geq 2$, it holds $\bar{R}_b(h) \leq M$ for any $h \in [0, M]$. To show this, observe that

$$\bar{R}_b(h) \leq M \iff M - \frac{1}{M}h^2 + \left(\frac{h}{M}\right)^d (M - h) \leq M$$

$$\iff \left(\frac{h}{M}\right)^d (M - h) \leq \frac{1}{M}h^2$$

$$\iff \left(\frac{h}{M}\right)^{d-1} (M - h) < h.$$

To show the above holds for all $h \in [0, M]$, we rescale $h$ by setting $h = \rho M$, where $\rho \in [0, 1]$. Then, the above becomes
$$\left(\frac{\rho M}{M}\right)^{d-1} (M - \rho M) < \rho M,$$

i.e. $\rho^{d-2}(1 - \rho) < 1$, which clearly holds for all $\rho \in [0, 1]$ when $d \geq 2$.

We finally verify that maximum of $R_r(x)$ is attained in the interior of $[0, M]$. First note that $R_r(0) = 0$ and $R_r(M) = 1$, so it suffices to show that $\max_{x \in [0,M]} R_r(x) > 1$. To this aim, note that $R_r(M - h) - R_b(M - h) = (r - b)h^d$, and the proof follows. $\quad\square$

It will be convenient for the proof to only consider policies represented by trees where the node prices never change after $\frac{T}{2}$.

**Lemma 18** ([9]). For any markdown policy $\mathbb{A}$, there is a policy $\mathbb{B}$ which makes no price change after $\frac{T}{2}$ such that $\mathrm{Reg}(\mathbb{B}, R) \leq 2 \cdot \mathrm{Reg}(\mathbb{A}, R)$ for any Lipschitz reward function $R$.

Thus by losing a constant factor in regret, we may consider only policies with no price changes after $\frac{T}{2}$. We first construct an adaptive classifier $(\Omega', x', f')$ as follows. With some foresight choose $h = T^{-\frac{1}{2d+2}}$. Let $\Omega' = \{v \in \Omega : d(v) \leq \frac{T}{4}, \text{ and } x(v) \geq 1 - h\}$, and $x' = x|_{\Omega'}$. Define $f : \Omega' \to \{R, B\}$ as

$$f'(\ell) = \begin{cases} B, & \text{if } x(\ell) > 1 - h, \\ R, & \text{else.} \end{cases}$$

Recall that $(\Omega', x', f')$ is $(\alpha, \beta)$-confident if $\mathbb{P}_b(f^{-1}(R)) \leq \alpha$ and $\mathbb{P}_r(f^{-1}(B)) \geq \beta$. We first show that if $\pi$ has the target regret, then $(\Omega', x', f')$ has to be $(\frac{1}{3}, \frac{2}{3})$-confident. Formally, we have the following lemma.

**Lemma 19.** If $\mathrm{Reg}(\pi, \mathcal{F}) \leq \frac{1}{4} T^{\frac{d}{d+1}}$, then $(\Omega', x', f')$ is $(\frac{1}{3}, \frac{2}{3})$-confident.

*Proof.* For contradiction, suppose $(\Omega', x', f')$ is not $(\frac{1}{3}, \frac{2}{3})$-confident. Then there are two cases. Let $N(a, b)$ be the number of rounds the policy selects prices from the interval $[a, b]$.

- First suppose $\mathbb{P}_R[f^{-1}(B)] = \mathbb{P}_R[x\left(\frac{T}{4}\right) > 1 - h] > \frac{1}{3}$, then $\mathrm{Reg}(\pi, R) \geq \frac{T}{4} \cdot \frac{1}{3} = \frac{T}{12} > \frac{1}{4} T^{\frac{d}{d+1}}$, a contradiction.

- Now suppose $\mathbb{P}_B[f(L) = R] = \mathbb{P}_B[x\left(\frac{T}{4}\right) \leq 1 - h] > \frac{1}{3}$. Note that $R'_B(1) = 0$, so at least $h^2$ regret is incurred in each remaining round. Since there are $\frac{T}{4}$ rounds remaining, the total regret in this case is at least $h^2 \cdot \frac{T}{4} = \frac{1}{4} T^{\frac{d}{d+1}}$, a contradiction.

$\square$

By Theorem 11 and noting that $\mathrm{KL}(R_r(x), R_b(x)) \leq h^{2d}$, we immediately obtain the following. Recall that $D(\ell)$ is the level of a leaf $\ell \in L(\Omega')$.

**Lemma 20.** Suppose $(\Omega', x', f')$ is $(\frac{1}{3}, \frac{2}{3})$-confident, then $\mathbb{E}_R[D] = \Omega(h^{-2d})$.

Note that the regret per round in $[1 - h, 1]$ under $D_r$ is $\Omega(1)$, thus for any policy $\pi$ with $O(T^{\frac{d}{d+1}})$ regret, by Lemma 19 and 20,

$$\mathrm{Reg}(\pi, R) \geq \mathbb{E}_R[N(1 - h, h)] \cdot \Omega(1) \geq h^{-2d} \cdot \Omega(1) = \Omega(T^{\frac{d}{d+1}}),$$

and Theorem 4 follows.

### C.4 Infinite-Dimensional Family

We next show Theorem 6. The proof uses similar idea as in the lower bound proof in [9]. However, for each $s \geq 2$ we need to construct an $s$-sensitive family of demand functions.

We consider the following $s$-sensitive family of unimodal reward curves. With some foresight, choose $h = T^{-\frac{1}{3s+1}}$ and for simplicity assume $m := \frac{1}{h}$ is an even integral. Consider the following S-shaped curves (or *S curves*) and bow-shaped curves (or *B curves*) as defined as follows. First Define a decreasing sequence $x_i = 1 - (2i - 1)h$ for $i = 1, ..., \frac{m}{2}$. Each pair of curves $B_i, S_i$ are defined in the interval $[x_i - h, x_i + h]$ (see Figure 4) such that

$$B_i(x_i + \xi) = y_i - |\xi|^s, \quad \forall \xi \in [-h, h],$$

and

$$S_i(x_i + \xi) = \begin{cases} y_i + |\xi|^s, & \text{if } \xi \in [-h, 0], \\ y_i - |\xi|^s, & \text{if } \xi \in [0, h], \end{cases}$$

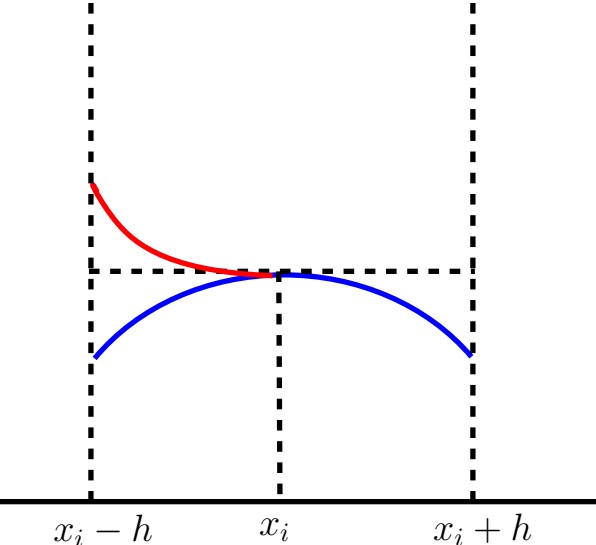

Figure 4: Bow-shaped (blue) and S-shaped (red) reward curves

where $y_i = \frac{1}{2} + 2ih^s$.

Now we construct the reward function $R_i$ using these gadgets. Loosely, for $i = 1, ..., \frac{m}{2}$, $R_i$ is a concatenation of $(i-1)$ consecutive S-curves from right to left, followed by one B curve, and finally a curve extending downwards the left portion of $B$ until reaching the $x$-axis. Formally for any $i = 1, ..., \frac{m}{2}$,

$$R_i(x) = \begin{cases} S_j(x), & \text{if } x \in [x_j - h, x_j + h] \text{ for } j \leq i - 1, \\ B_i(x), & \text{if } x \in [x_i - h, x_i + h], \\ sh^{s-1}x + \left(y_i - h^s - sh^{s-1}(x_i - h)\right) & \text{if } x \leq x_i - h. \end{cases}$$

Finally, we need a special reward function $R_0$, that consists only of S-curves on $[\frac{1}{2}, 1]$, and extends upwards when the prices moves below $\frac{1}{2}$, analogous to the construction to the roof curves in the lower bound proof in [9]. Formally,

$$R_0(x) = \begin{cases} R_{\frac{m}{2}}(x), & \text{if } x \geq 1/2, \\ y_{\frac{m}{2}} + (x_{\frac{m}{2}} - x)^s, & \text{if } x \in [0, 1/2]. \end{cases}$$

The lower bound is again showed using Theorem 11. Our proof is similar to that of Theorem 3 in [9] and we will only sketch the proof. Fix an arbitrary $i \in [m]$ and consider $R_i$ We first show that if $\pi$ has $O(T^{\frac{2s+1}{3s+1}})$ regret, then $\pi$ is able to distinguish between $R_0$ and $R_i$ with constant confidence level (more precisely, a suitable adaptive classifier is $(\Omega(1), O(1))$-confident). Suppose $R_i$ is the true reward function and $\pi$ has an $\Omega(1)$ probability of mistakenly returning $R_0$ as the true curve. In other words, with $\Omega(1)$ probability, reduces the price below $x_i - h$, incurring an $\Omega(h^s)$ regret in each remaining round. This leads to an $\Omega(h^s T) = \Omega(T^{\frac{2s+1}{3s+1}})$ cumulative regret, a contradiction.

We next show that the expected number of rounds in $[x_i - h, x_i + h]$ is $\Omega(h^{-2s})$. Note that $R_0$ and $R_i$ are identical on $[x_i, 1]$, where no progress can be made towards distinguishing between $R_i$ and $R_0$. Since the maximum KL divergence on $[x_i - h, x_i + h]$ is $O(h^{2s})$, by Theorem 11, $\Omega(h^{-2s})$ samples are necessary in expectation assuming the policy $\pi$ is able to distinguish between these two reward functions.

Finally, since the above argument holds for every $i \in [m]$, we have

$$\text{Reg}(\pi, R_0) \geq \Omega(h^{-2s}) \cdot m = \Omega(h^{-1-2s}) = T^{\frac{2s+1}{3s+1}},$$

and the proof follows.