# OpenReview forum: "Dynamic Pricing with Monotonicity Constraint under Unknown Parametric Demand Model"
_NeurIPS.cc/2022/Conference — NeurIPS 2022 Accept_

### Official Review · Reviewer_wcPP · 2022-07-10

**Rating:** 5
**Confidence:** 3
**Soundness:** 3 good
**Presentation:** 3 good
**Contribution:** 3 good

**Summary:**

This paper considers infinite-arm MAB with markdown constraints. Since existing results have shown that $T^{3/4}$ is optimal under minimal assumptions, the authors turn to study how to further improve the regret bound by assuming additional assumptions for the demand functions. To this end, they introduce a general complexity notion called *markdown dimension*. Using this notion, they not only show that a better regret bound is possible under certain complexities, but there still exists a separation between MAB with constraints and without markdown constraints.

**Questions:**

Several improvements are possible based on the current status of this paper.
- It might need more intuitions or explanations when introducing the new notion *markdown dimension*. In fact, the reviewer thinks that it would be much better to first consider some special cases and then introduce the general notion based on the intuitions from the special cases. This will help readers appreciate the contributions.
- The current version does not contain any experiments. The reviewer thinks that this is even more necessary for this paper since the goal is to show some improvement under each particular function class. Thus, it would be good to see improvements from experiments, at least some simple numerical experiments.
- Some possible typos: (1) the first line on Page 2, extra 'this bound'? (2) Proposition 3, b should be a?

**Limitations:**

Yes.  The authors adequately addressed the limitations

**Strengths And Weaknesses:**

**Strengths**
- A comprehensive study of MAB with markdown constraints
- Matching upper and lower bounds

**Weaknesses**
- No experimental results
- Writing needs improvement

---

> ### Author Response · Authors · 2022-08-09
> **Author Response**
>
> **We first apologize for submitting the entire rebuttal as one pdf file to the paper submission portal, which may have caused troubles for reviewers to notice. Our original rebuttal submission includes (1) a combined response letter in pdf, and (2) a revised paper, with new contents in blue color. While you can still access the pdf version of rebuttal file, here we copy and paste our response from that file:**
>
> To the entire review team:
>
> Thank you for a positive and thoughtful evaluation of our paper, we were grateful to receive constructive comments from the reviewers. In response, we have made a number of substantial improvements to the paper. We summarize the major changes below.
>
> 1. *Explanation of Markdown Dimension.* Reviewers ZpZo and WcPP believe that the explanation of our key concept, the *markdown dimension*, can be improved. We added one subsection (Subsection 2.2 in the revised version) that analyzes the nice properties of the linear demand family (“robustness under noisy demands”) and used it to motivate more general markdown dimensions.
>
> 2. *Monotonicity of Prices in Algorithm 1.* We thank reviewers ZpZo and CjUD for pointing out that in Algorithm 1 the prices are not guaranteed to be decreasing. This can be fixed by a slight modification. We added Line 12 in Algorithm 1, which ensures monotonicity: it compares the estimated optimal price $\tilde p_{j+1} = \arg \max ( p^*(\theta) : |\theta − \theta^*| \leq w_{j+1} )$ (called the “raw” price), with the previous price $p_j$.
> We show, however, that such modification does not lead to significant change in the analysis. In fact, we show that with $1 − O(T^{−2})$ probability, we have $\tilde p_{j+1} \leq p_j$. Thus, by losing an $O(T ) · O(T^{−2}) = o(1)$ factor in the regret (due to the low probability event), our previous analysis goes through, that is, it suffices to just analyze the regret for the price sequence $\{\tilde p_{j+1}\}$. We have revised the analysis for $d = 0$ accordingly in the supplementary materials.
>
> 3. *Missing Proofs.* Reviewer CjUD pointed out that the proofs of Propositions 1,2,3,4 are missing. While the proofs are actually fairly straightforward, we agree with the reviewers that adding them would make it easier for readers to digest the definition of markdown dimension, as well as improving the completeness of the paper. We have therefore added their proofs in the revised supplementary materials.
>
> 4. *Experiments.* Some reviewer suggested that numerical simulations would significantly improve this work. This is a good suggestion, but unfortunately, due to the large volume of other issues to address, we were not able to prioritize the experiments in the rebuttal period, but we will add a comprehensive experiment in the camera-ready version if this work is accepted.
>
> 5. *Discussion of Limitations.* Reviewer ZpZo mentioned we did not discuss the limitations of our work. In the revised version, we explicitly pointed out the following limitations of our work in various places of the main body:
> - As reviewer ZoQT pointed out, all 3 policies we presented require the knowledge of d, and these policies look quite different. We leave it open whether there exists a "universal" policy that handles all these regimes.
> - As reviewer ZpZo pointed out, our Lipschitz assumption on p^* (Assumption 5) is somewhat indirect and may sometimes be hard to verify. While we believe it can be implied by some "direct" assumptions on the parametrization, at this moment we are not sure what these direct assumptions should be. To be safe, please let us keep Assumption 5 and add a remark that we leave it open whether Assumption 5 can be relaxed/replaced.

---

### Official Review · Reviewer_ZpZo · 2022-07-10

**Rating:** 7
**Confidence:** 3
**Soundness:** 3 good
**Presentation:** 2 fair
**Contribution:** 4 excellent

**Summary:**

This work studies a single-product pricing problem under a monotonic/markdown price constraint. In this work, the authors introduce a "markdown dimension" $d$ that indicates the hardness of a demand curve to be learnd from customers' feedback. For each of the following cases: $d=0$, $d\in\mathbb{Z}^+$, $d=\infty$, the authors propose a pricing algorithm with provable expected regret guarantees. Specifically, these bounds are optimal for $s=2$ as the authors also prove matching lower bounds. In conclusion, this work improves existing results in related literatures and introduce a new method of determining the hardness of a pricing problem.

**Questions:**

1, By the end of the first paragraph of Section 2: What do you mean by "almost surely"? As T is finite, an "almost surely" constraint means no violations at all, right?

2, In the second paragraph of Section 2: It is better to define a policy $\pi$ formally, as it is time-and-history dependent.

3, Assumption 1: Why the optimal price can be assumed to be an interior point instead of on the boundary, without loss of generality?

4, The first equation on the top of Page 4: With only differentiable assumption, there does not necessarily exist such a second-order derivatives. Notice that a twice-differentiable assumption is in Assumption 4. Therefore, I suggest the authors to state Assumption 1 more clearly.

5, One line before Assumption 5: I suggest the authors to formally define the $p^*$ function in formulas.

6, Assumption 5: $p^*$ is a function mapping the parameter of D to the optimal price of D. On the one hand, the property of $p^*$ is dependent on that of $D$, especially on how a $\theta$ determines a $D(p; \theta)$. Also, $p^*$ is not an explicit function under most circumstance, but $D$ is obviously more tractible. So I think it is not a proper way to make assumptions on $p^*$ directly.

7, Proposition 4: Is this easy to be proved?

8, The last paragraph on Page 6: What is $C_s$? (I guess it is the s-th coefficient of the $p^s$ term of R's polynomial approximation, but not sure if I am right.) Besides, this inequality is not always true as you neglect the Lagrangian remainders.

9, The first line on Page 7: It is NOT what [10] have stated. Their definition on $\alpha$ (albeit the $s$ here) is a Lipschitz exponent instead of a sensitive parameter. In other words, they aim at upper bound the difference while sensitivity is a lower bound of difference. This might be caused by a mistake in stating Definition 10 (i.e., $-C_6$ serves as a lower bound instead of an upper bound), but I got really confused about its name "sensitivity". Intuitively, a larger sensitivity would cause a larger fluctuation on reward (or demand) while changing the price, but a larger $s$ would definitely "smoothen" the demand curve and reduce the fluctuation caused by price changing. I am very curious why the authors did not call it a "smoothness"? Maybe there exist some causes in depth, and I am also not sure if this would affect the soundness.

**10, Algorithm 1 Line 11: How to ensure $p_{j+1} \leq p_j$ as $\hat{\theta}_j$ is a random variable?

11, In Theorem 3: It might be better to rewrite the expression for $s=2$ and $s>2$, which would make both of them concise and informative.

12, A few typos I have noticed:

    (1) Section 1.1 line 7: "a efficient" --> "an efficient"
    (2) Page 3 line 6: "For for" --> "For"
    (3) Page 4 line 6 from bottom: "In words" --> "In other words"
    ......

I recommend the authors to carefully proofread the paper and fix all typos and syntax errors.

**Limitations:**

No discussions found in this paper. I recommend the authors to discuss the limitations of their work at the end of the main pages. There seems no negative social impact or ethic issue, and I also encourage the authors to discuss them in their Appendix.

**Strengths And Weaknesses:**

Strengths:

1, The problem setting and constraints are practical: customers are indeed more sensitive to price raising than discounting.

2, The new concept on "markdown dimensionality" is new in the field of pricing research. Also, it helps determine the hardness of pricing problems.

3, Most of the regret upper bounds of the algorithms this paper proposes are proved to be optimal (up to log or loglog factors) by the authors.


Weaknesses:

1, The writing and organization of this paper is not good. In specific, there are many obvious typos and syntax errors even in the most important definitions. For instance, see Definition 10: I really suspect that the authors made a mistake in this definition, especially (c): all their applications of this definition point at a smoothness, i.e., $\exists C>0$ such that $-C\leq R^{(s)}(\cdot)<0$.

Based on this definition, a sensitivity can only be used to lower bound the price perturbation instead of upper-bounding it. E.g., for s=2, it is somewhat a strong convexity.

2, Many key definitions and statements are ambiguous or misleading. For example, in Definition 8: Is there a definition of parameterization? and How does $\theta$ relate to the following conditions in (2)? Notice that $\Phi^{-1}_P(y)$ is not necessary a constant unless F is d-identifiable. Also, what does "it" in (1) refer to?

3, For key definitions, propositions and assumptions, there are lack of explanations or insights that help the readers to understand. For example, in Definition 2: I had spent a long time until I realized that $p^*$ is the lowest best price and $p^* (R)$ is an argmax oracle. In contrast, there are redundant explanations on trivial facts. For example, the matrix-form equation under Assmption 5 is not necessary and it can be clearly described just by two equations $D = \sum_{j=0}^d\theta_j p^j$ and $\theta = V_p^{-1}y$.

4, The authors did not show the "markdown dimension" of a variety of common distributions except for the $0$-dimensional ones. In other words, there exists no example on $d\geq1$ and $d=\infty$, which qualifies the application of most of their theoretical results.

5, This is very important to get the authors' notice:

    Notice that the authors added more contents in their main pages in the supplementary materials. This is unfair to the other authors and their submissions as this actually breaks the 9-page-limit rule (although it is in the supplementary materials). The authors should separate their main pages with the proof details of upper and lower bounds and put them into an Appendix. Besides, the format of this paper seems like a "preprint" instead of a "submission" as an option in the Latex template.



From my point of view, this paper has strong technical contributions, but the authors convey these contributions in a careless way. Not sure if it is suitable for publishing in Neurips. I'll tentatively give a 6 in advocate of their theoretical results and see what the other reviewers would say.

---

> ### Comment · Reviewer_ZpZo · 2022-08-08
> **Post-rebuttal feedback**
>
> **Notice that the authors submitted their rebuttal to the paper submission portal, which I guess is very hard for reviewers to notice.**
>
> The paper has been improved a lot from the previous version, and the authors has made sufficient clarifications or explanations to most of my questions. However, it seems like the authors did not notice those issues I pointed out in the "Strength and Weakness" part, and some of my questions there were not answered.
>
> Overall, this paper is contributing to research on dynamic pricing as they studied a practical "markdown pricing" setting and propose a "markdown dimension" notion that would facilitate further research. From the perspective of technical contribution, I tend to put aside some minor controversies and increase my score to 7.

---

> > ### Author Response · Authors · 2022-08-09
> > **Thank you for informing us of mishandling the rebuttal portal**
> >
> > We have copied our response from the pdf file to the other reviewers' "official comment" section.

---

### Official Review · Reviewer_CjUD · 2022-07-10

**Rating:** 5
**Confidence:** 4
**Soundness:** 2 fair
**Presentation:** 3 good
**Contribution:** 3 good

**Summary:**

The paper considers the monotone (markdown) price constraint in the dynamic pricing problem, where the demand has a parametric form in price.

The paper introduces a concept, the markdown dimension $d$, which measures the complexity of the parametric family.
When the dimension $d=0, 1 \le d < \infty, d= \infty$, the paper proposes algorithms achieving the regret $O(\log^2 T), O(T^{d/(d+1)}), O(T^{(2s+1)/(3s+1)})$. Here, $s$ measures the degree of smoothness of revenue function at the optimal price. Also, the matching lower bounds are provided for each case.


**Questions:**

The major question is about Algorithm 1 (on page 7). Is there any step in Algorithm 1 to make sure the prices $p_j$ are decreasing? More precisely, in phase $j$ the parameter $\theta_j$ is estimated, and the estimation may be inaccurate with a probability. The conservative price $p_{j+1}$ (line 11) obtained by the inaccurate estimation may be larger than $p_j$. After checking the algorithm and proof, I have not found any steps to prevent this.

The regret for the parametric case when $d=\infty$, does not match the regret O(T^{3/4}) for the nonparametric unimodal case. So after imposing the parametric assumption (when $d>3$), the regret may be worse. Yet it is shown that it matches the lower bound. Does it make the definition of $d$ unnatural?

Also, I have not found the proof for Proposition 1,2,3,4 in the paper or appendix.

Some typos: in Proposition 3, $b$ should be $a$. In the 2nd line of Theorem 3, there’s two "policy".


**Strengths And Weaknesses:**

Originality: It’s interesting to consider the monotonicity constraint in the parametric function. The paper shows some new results which are different from the nonparametric case. However, I find the definition of markdown dimension $d$ a little unintuitive, and it is hard to see if the concept can be applied to other applications.

Quality: It’s technically sound, except for Algorithm 1 and Theorem 1. See Questions.

Clarity: The paper is written clearly.

There are three main results in the paper.

First, for the simple case when the dimension $d=0$, the paper shows the minimax regret bound $O(\log^2 T)$ which is different from the unconstraint pricing case $O(\log T)$. The result is not surprising but still adds to the literature. In this case, there’s a one-to-one mapping from the parameter to the realized demand. Every price is informative and there’s no trade-off between learning and earning. My major concern is whether the prices in the algorithm are actually decreasing. See Questions for details.

Second, when $d >=1$, the paper shows the minimax regret bound $O(T^{d/(d+1)})$ which are new results. I think that’s the major contribution of the paper. The learning and earning tradeoff shows up in the hyperparameter $h$, i.e., the smallest magnitude that the price decreases between consecutive periods. Setting $h=O(T^{-1/(d+1)})$, the regret will be $O(T^{d/(d+1)})$. My question is how to connect the parametric result with the existing nonparametric result. See Questions for details.

Third, when the function is nonparametric, the paper assumes a higher order of smoothness and proposes an algorithm (Algorithm 3). I think the contribution of this part is marginal because the generalization from Lipschitz continuous to higher-order smoothness is explored by the literature.

---

> ### Author Response · Authors · 2022-08-09
> **Author Response**
>
> **We first apologize for submitting the entire rebuttal as one pdf file to the paper submission portal, which may have caused troubles for reviewers to notice. Our original rebuttal submission includes (1) a combined response letter in pdf, and (2) a revised paper, with new contents in blue color. While you can still access the pdf version of rebuttal file, here we copy and paste our response from that file:**
>
> To the entire review team:
>
> Thank you for a positive and thoughtful evaluation of our paper, we were grateful to receive constructive comments from the reviewers. In response, we have made a number of substantial improvements to the paper. We summarize the major changes below.
>
> 1. *Explanation of Markdown Dimension.* Reviewers ZpZo and WcPP believe that the explanation of our key concept, the *markdown dimension*, can be improved. We added one subsection (Subsection 2.2 in the revised version) that analyzes the nice properties of the linear demand family (“robustness under noisy demands”) and used it to motivate more general markdown dimensions.
>
> 2. *Monotonicity of Prices in Algorithm 1.* We thank reviewers ZpZo and CjUD for pointing out that in Algorithm 1 the prices are not guaranteed to be decreasing. This can be fixed by a slight modification. We added Line 12 in Algorithm 1, which ensures monotonicity: it compares the estimated optimal price $\tilde p_{j+1} = \arg \max ( p^*(\theta) : |\theta − \theta^*| \leq w_{j+1} )$ (called the “raw” price), with the previous price $p_j$.
> We show, however, that such modification does not lead to significant change in the analysis. In fact, we show that with $1 − O(T^{−2})$ probability, we have $\tilde p_{j+1} \leq p_j$. Thus, by losing an $O(T ) · O(T^{−2}) = o(1)$ factor in the regret (due to the low probability event), our previous analysis goes through, that is, it suffices to just analyze the regret for the price sequence $\{\tilde p_{j+1}\}$. We have revised the analysis for $d = 0$ accordingly in the supplementary materials.
>
> 3. *Missing Proofs.* Reviewer CjUD pointed out that the proofs of Propositions 1,2,3,4 are missing. While the proofs are actually fairly straightforward, we agree with the reviewers that adding them would make it easier for readers to digest the definition of markdown dimension, as well as improving the completeness of the paper. We have therefore added their proofs in the revised supplementary materials.
>
> 4. *Experiments.* Some reviewer suggested that numerical simulations would significantly improve this work. This is a good suggestion, but unfortunately, due to the large volume of other issues to address, we were not able to prioritize the experiments in the rebuttal period, but we will add a comprehensive experiment in the camera-ready version if this work is accepted.
>
> 5. *Discussion of Limitations.* Reviewer ZpZo mentioned we did not discuss the limitations of our work. In the revised version, we explicitly pointed out the following limitations of our work in various places of the main body:
> - As reviewer ZoQT pointed out, all 3 policies we presented require the knowledge of d, and these policies look quite different. We leave it open whether there exists a "universal" policy that handles all these regimes.
> - As reviewer ZpZo pointed out, our Lipschitz assumption on p^* (Assumption 5) is somewhat indirect and may sometimes be hard to verify. While we believe it can be implied by some "direct" assumptions on the parametrization, at this moment we are not sure what these direct assumptions should be. To be safe, please let us keep Assumption 5 and add a remark that we leave it open whether Assumption 5 can be relaxed/replaced.

---

### Official Review · Reviewer_Vvni · 2022-07-11

**Rating:** 5
**Confidence:** 4
**Soundness:** 2 fair
**Presentation:** 3 good
**Contribution:** 2 fair

**Summary:**

The paper deals with learning pricing with markdown constraints, that is that subsequent prices must be non decreasing. The authors investigate this problem in the case where the demand is a parametric function of the price, with known form but unknown values of the parameters - in this case, a tradeoff must be found between exploring to accurately learn the parameters of the model and losing a lot due to exploration (and the markdown constraint).

 The main contributions of the paper are (i) the introduction of "markdown dimension", which quantifies the how difficult it is to learn the parameters of the model from data (ii) an algorithm that balances exploration with good performance (under the concept of pessimism in the face of uncertainty) and
(iii) an algorithm that matches the regret lower bound when the markdown dimension of the problem is infinity (in which case no algorithms with sublinear regret exist).


**Questions:**

1. Related to the weaknesses above - please illustrate, with some concrete examples, why the setting is relevant.
2. It would be nice to provide numerical results with real datasets illustrating the performance of the proposed algorithms by fitting/assuming different kinds of models for the data.

**Limitations:**

No potential negative societal impact foreseen.

**Strengths And Weaknesses:**

Strengths
1. The analysis is nice and rigorous.
2. From a mathematical point of view, introduction of the markdown dimension to characterize the difficulty of the problem is an intersting idea.
3. Also, the result that knowledge of the demand model can lead to improvements in the regret (and the algorithms to achieve this) can be a good addition to the related literature of pricing with markdown constraints.

Weaknesses
1. The main weakness is that th setting is not convincingly motivated:  Since the markdown constraint is motivated by pricing problems in markets of some sort, I am not sure it is reasonable to assume that the form of demand (essentially how the market will react to the price) is known. There is no concrete example where a parametric form of the demand model is reasonable/accurate and no  numerical results to illustrate the performance of the proposed algorithms vs standard algorithms in practical problem settings.

---

> ### Author Response · Authors · 2022-08-09
> **Author Response**
>
> **We first apologize for submitting the entire rebuttal as one pdf file to the paper submission portal, which may have caused troubles for reviewers to notice. Our original rebuttal submission includes (1) a combined response letter in pdf, and (2) a revised paper, with new contents in blue color. While you can still access the pdf version of rebuttal file, here we copy and paste our response from that file:**
>
> To the entire review team:
>
> Thank you for a positive and thoughtful evaluation of our paper, we were grateful to receive constructive comments from the reviewers. In response, we have made a number of substantial improvements to the paper. We summarize the major changes below.
>
> 1. *Explanation of Markdown Dimension.* Reviewers ZpZo and WcPP believe that the explanation of our key concept, the *markdown dimension*, can be improved. We added one subsection (Subsection 2.2 in the revised version) that analyzes the nice properties of the linear demand family (“robustness under noisy demands”) and used it to motivate more general markdown dimensions.
>
> 2. *Monotonicity of Prices in Algorithm 1.* We thank reviewers ZpZo and CjUD for pointing out that in Algorithm 1 the prices are not guaranteed to be decreasing. This can be fixed by a slight modification. We added Line 12 in Algorithm 1, which ensures monotonicity: it compares the estimated optimal price $\tilde p_{j+1} = \arg \max ( p^*(\theta) : |\theta − \theta^*| \leq w_{j+1} )$ (called the “raw” price), with the previous price $p_j$.
> We show, however, that such modification does not lead to significant change in the analysis. In fact, we show that with $1 − O(T^{−2})$ probability, we have $\tilde p_{j+1} \leq p_j$. Thus, by losing an $O(T ) · O(T^{−2}) = o(1)$ factor in the regret (due to the low probability event), our previous analysis goes through, that is, it suffices to just analyze the regret for the price sequence $\{\tilde p_{j+1}\}$. We have revised the analysis for $d = 0$ accordingly in the supplementary materials.
>
> 3. *Missing Proofs.* Reviewer CjUD pointed out that the proofs of Propositions 1,2,3,4 are missing. While the proofs are actually fairly straightforward, we agree with the reviewers that adding them would make it easier for readers to digest the definition of markdown dimension, as well as improving the completeness of the paper. We have therefore added their proofs in the revised supplementary materials.
>
> 4. *Experiments.* Some reviewer suggested that numerical simulations would significantly improve this work. This is a good suggestion, but unfortunately, due to the large volume of other issues to address, we were not able to prioritize the experiments in the rebuttal period, but we will add a comprehensive experiment in the camera-ready version if this work is accepted.
>
> 5. *Discussion of Limitations.* Reviewer ZpZo mentioned we did not discuss the limitations of our work. In the revised version, we explicitly pointed out the following limitations of our work in various places of the main body:
> - As reviewer ZoQT pointed out, all 3 policies we presented require the knowledge of d, and these policies look quite different. We leave it open whether there exists a "universal" policy that handles all these regimes.
> - As reviewer ZpZo pointed out, our Lipschitz assumption on p^* (Assumption 5) is somewhat indirect and may sometimes be hard to verify. While we believe it can be implied by some "direct" assumptions on the parametrization, at this moment we are not sure what these direct assumptions should be. To be safe, please let us keep Assumption 5 and add a remark that we leave it open whether Assumption 5 can be relaxed/replaced.

---

### Official Review · Reviewer_ZoQT · 2022-07-11

**Rating:** 6
**Confidence:** 3
**Soundness:** 4 excellent
**Presentation:** 3 good
**Contribution:** 3 good

**Summary:**

This paper studies the dynamic (monopolistic) pricing problem under “markdown constraints” which requires pricing decisions to be nonincreasing. The decision maker does not know the form of the demand function that stays constant over time, receives bandit feedback for decisions made, and aims to minimize cumulative regret compared to the maximum achievable revenue under the optimal price. The paper first proposes a categorization for demand functions called markdown dimensionality to describe the complexity of demand functions in the context of markdown pricing. Then, the paper presents dynamic pricing algorithms for each category, respectively, as well as matching regret lower bounds demonstrating the proposed algorithms are tight.

**Questions:**

The two key characteristics for demand functions in this paper are the markdown dimension and sensitivity. However, I noticed that there is no discussion on how these parameters collectively affect regret. For example, in Theorem 1 and 2, is there any intuition behind why there is a discontinuous “jump” in regret for $s=2$ for 0 markdown dimension, but for finite markdown dimension such a “jump” disappears?

In Algorithm 2 and Theorem 3, how should the decision maker choose $h$ and the phase lengths $T_1 ….T_m$?


**Strengths And Weaknesses:**

Strength:
The paper is well-written, and to the best of my knowledge, the key contributions regarding 1. defining markdown dimensions; 2. developing algorithms to achieve near optimal regret for each markdown dimension regime (i.e. 0, finite, and infinite markdown dimensions); and 3. presenting regret lower bounds for each regime, are novel. In my opinion, the paper presents a valuable framework to characterize the hardness of learning and dynamic monopolistic pricing under monotone constraints, and may lead to interesting research directions. I also think the paper positions itself well compared to existing papers for non-constrained dynamic pricing, as well as existing work on dynamic pricing with markdown constraints. The paper also presented a clear illustration for technical definitions such as the markdown dimension via concrete examples, and also for the proposed algorithms.

Weakness:
In my opinion, the main weakness of the paper is that individual algorithms are presented for different demand markdown-dimension regimes, meaning that the decision maker would need to know whether the supposedly unknown demand function’s markdown dimension is 0, finite or infinite, in order for a decision maker to deploy the corresponding algorithm. Also, in the finite, non-zero markdown dimension regime, the proposed Algorithm 2 requires the decision maker to know the underlying markdown dimension of demand. The paper would be much stronger if it can either propose a single “best of all worlds” algorithm that can achieve optimal regret under demands with any markdown dimensions while being agnostic to the regime, or analyze how the algorithms proposed in the papers would perform if the regime is misspecified. This suggestion may be beyond the scope of the paper, but perhaps it would be helpful to run some simulations to shed light on relevant aspects.

---

> ### Author Response · Authors · 2022-08-09
> **Author Response**
>
> **We first apologize for submitting the entire rebuttal as one pdf file to the paper submission portal, which may have caused troubles for reviewers to notice. Our original rebuttal submission includes (1) a combined response letter in pdf, and (2) a revised paper, with new contents in blue color. While you can still access the pdf version of rebuttal file, here we copy and paste our response from that file:**
>
> To the entire review team:
>
> Thank you for a positive and thoughtful evaluation of our paper, we were grateful to receive constructive comments from the reviewers. In response, we have made a number of substantial improvements to the paper. We summarize the major changes below.
>
> 1. *Explanation of Markdown Dimension.* Reviewers ZpZo and WcPP believe that the explanation of our key concept, the *markdown dimension*, can be improved. We added one subsection (Subsection 2.2 in the revised version) that analyzes the nice properties of the linear demand family (“robustness under noisy demands”) and used it to motivate more general markdown dimensions.
>
> 2. *Monotonicity of Prices in Algorithm 1.* We thank reviewers ZpZo and CjUD for pointing out that in Algorithm 1 the prices are not guaranteed to be decreasing. This can be fixed by a slight modification. We added Line 12 in Algorithm 1, which ensures monotonicity: it compares the estimated optimal price $\tilde p_{j+1} = \arg \max ( p^*(\theta) : |\theta − \theta^*| \leq w_{j+1} )$ (called the “raw” price), with the previous price $p_j$.
> We show, however, that such modification does not lead to significant change in the analysis. In fact, we show that with $1 − O(T^{−2})$ probability, we have $\tilde p_{j+1} \leq p_j$. Thus, by losing an $O(T ) · O(T^{−2}) = o(1)$ factor in the regret (due to the low probability event), our previous analysis goes through, that is, it suffices to just analyze the regret for the price sequence $\{\tilde p_{j+1}\}$. We have revised the analysis for $d = 0$ accordingly in the supplementary materials.
>
> 3. *Missing Proofs.* Reviewer CjUD pointed out that the proofs of Propositions 1,2,3,4 are missing. While the proofs are actually fairly straightforward, we agree with the reviewers that adding them would make it easier for readers to digest the definition of markdown dimension, as well as improving the completeness of the paper. We have therefore added their proofs in the revised supplementary materials.
>
> 4. *Experiments.* Some reviewer suggested that numerical simulations would significantly improve this work. This is a good suggestion, but unfortunately, due to the large volume of other issues to address, we were not able to prioritize the experiments in the rebuttal period, but we will add a comprehensive experiment in the camera-ready version if this work is accepted.
>
> 5. *Discussion of Limitations.* Reviewer ZpZo mentioned we did not discuss the limitations of our work. In the revised version, we explicitly pointed out the following limitations of our work in various places of the main body:
> - As reviewer ZoQT pointed out, all 3 policies we presented require the knowledge of d, and these policies look quite different. We leave it open whether there exists a "universal" policy that handles all these regimes.
> - As reviewer ZpZo pointed out, our Lipschitz assumption on p^* (Assumption 5) is somewhat indirect and may sometimes be hard to verify. While we believe it can be implied by some "direct" assumptions on the parametrization, at this moment we are not sure what these direct assumptions should be. To be safe, please let us keep Assumption 5 and add a remark that we leave it open whether Assumption 5 can be relaxed/replaced.

---

### Meta-Review · Area_Chair_pMZR · 2022-09-01

**Recommendation:** Accept
**Confidence:** Certain

**Metareview:**

This paper focuses on an interesting problem, dynamic pricing.

The paper brings conceptual new ideas (markdown dimension) and associated algorithms.

I have 2 concerns:
1) it would be better if the algorithms were adaptive to the aforementioned dimension.
2) it would have been better if the authors had followed the instructions (and especially not updated the rebuttal as the revised version).

Point 1 would be future work, while point 2 is ok since the pdf can still be found in the submission files.

As a consequence, I recommend acceptance




**Award:**

No

---

### Decision · Program_Chairs · 2022-09-14

Accept